# Human herpesvirus 6B glycoprotein B postfusion structure, vulnerability mapping, and receptor recognition

Chu Xie[1,☉], Xin-Yan Fang[2,3,☉], Yuan-Tao Liu[1,☉], Xian-Shu Tian[1,☉], Lan-Yi Zhong[1], Pei-Huang Wu[1], Hang Zhou[1], Peng-Lin Li[1], Yan-Lin Yang[1], Zi-Ying Jiang[1], Sen-Fang Sui[2,3,4,*], Zheng Liu[2,*], Mu-Sheng Zeng[1,*], Cong Sun[1,*]

1 State Key Laboratory of Oncology in South China, Guangdong Provincial Clinical Research Center for Cancer, Guangdong Key Laboratory of Nasopharyngeal Carcinoma Diagnosis and Therapy, Sun Yat-sen University Cancer Center, Guangzhou, China, 2 Cryo-electron Microscopy Center, Southern University of Science and Technology, Shenzhen, Guangdong, China, 3 Department of Biology, Southern University of Science and Technology, Shenzhen, Guangdong, China, 4 State Key Laboratory of Membrane Biology, Beijing Frontier Research Center for Biological Structures, Beijing Advanced Innovation Center for Structural Biology, School of Life Sciences, Tsinghua University, Beijing, China

☉ These authors contributed equally to this work.
* suncong@sysucc.org.cn (CS); zengmsh@sysucc.org.cn (M-SZ); liuz3@sustech.edu.cn (ZL); suisf@mail.tsinghua.edu.cn (S-FS)

## Abstract

Human herpesvirus 6B (HHV-6B), a β-herpesvirus that significantly threatens immunocompromised individuals, currently lacks targeted antiviral therapies or vaccines. Glycoprotein B (gB), the primary mediator of membrane fusion during viral entry, is a key target for neutralizing antibody (nAb) and vaccine development. In this study, we determined a 2.8 Å cryo-EM structure of the HHV-6B gB ectodomain in its postfusion conformation, unveiling unique N-terminal features and resolving the furin site for the first time in herpesviruses. Comparative analyses highlighted similarities between HHV-6B gB and gB from human cytomegalovirus (HCMV) and Epstein-Barr virus (EBV), mapping conserved residues across herpesviruses. Cross-binding assays indicated minimal cross-epitope recognition by nAbs from other herpesviruses, while several potential vulnerable sites on HHV-6B gB were identified. These insights advance our understanding of HHV-6B infection mechanisms and support future development of antibodies or vaccines targeting gB.

## Author summary

Human herpesvirus 6B (HHV-6B) is a widespread β-herpesvirus that establishes lifelong latency and can reactivate, causing severe complications in immunocompromised individuals. Despite its clinical significance, there is limited understanding of its infection mechanisms and neutralization vulnerabilities. This study presents the first high-resolution cryo-EM structure of HHV-6B glycoprotein B

**Data availability statement:** The cryo-EM structure and density map of HHV-6B gB are accessible in PDB under accession number 9JLI and EMDB under accession number EMD-61589. Materials and plasmids can be requested from duwt@sysucc.org.cn. Raw experimental data supporting key findings (e.g., immunoprecipitation assays, SDS-PAGE results) have been deposited in the Research Data Deposit platform (www.researchda-ta.org.cn) under the accession number RDDB2025447512.

**Funding:** This work was supported by grants from the National Key Research and Development Program of China (2022YFC3400900 – M.S.Z.), National Natural Science Foundation of China (82030046 – M.S.Z., 82402614 – C.S., 82070329 and 32241028 – Z.L.), Postdoctoral Fellowship Program of CPSF (GZB20230886 – C.S.), China Postdoctoral Science Foundation (2023M743998, 2024T171080 – C.S.), the Fundamental Research Funds for the Central Universities (84000-31610027 – C.S.), the Fundamental Research Funds for the Central Universities, Sun Yat-sen University (24qnpy278 – C.S.). The funders had no role in study design, data collection and analysis, decision to publish, or preparation of the manuscript.

**Competing interests:** The authors have declared that no competing interests exist.

(gB), a key fusion protein essential for viral entry, and unveils its unique structural features. Comparative analyses identified conserved and distinct regions of gB across herpesviruses and highlighted potential vulnerabilities of HHV-6B gB. Additionally, the study reveals structural insights into gB's interaction with the receptor nectin-2, advancing our understanding of HHV-6B pathogenesis. This work paves the way for targeted interventions against HHV-6B infection and related diseases.

## Introduction

Herpesviruses are a large family of viruses that cause a wide range of diseases and are classified into three subfamilies: alpha, beta, and gamma. They establish lifelong infections with cycles of latency and reactivation. Human herpesvirus 6 (HHV-6) belongs to the beta-herpesvirus subfamily and is divided into two species, HHV-6A and HHV-6B, which share high sequence homology but differ significantly in epidemiology, pathogenesis, and clinical significance [1]. While HHV-6A is less characterized, HHV-6B has great clinical significance due to its widespread infection, affecting over 90% of the human population and the contributing to substantial morbidity [2]. HHV-6B infection usually occurs within the first two years of life, often with mild symptoms before the virus entering latency. However, in some cases, it can lead to severe conditions such as acute pediatric encephalopathy [3,4]. HHV-6B reactivation is a major concern for immunocompromised individuals, particularly transplant recipients. Approximately half of allogeneic hematopoietic cell transplantation (HCT) recipients experience HHV-6B reactivation, which is linked to a wide range of complications and increased mortality [5]. HHV-6B is the leading cause of infectious encephalitis after HCT, with higher viral DNA loads in bronchoalveolar lavage fluid strongly predicting respiratory failure and overall mortality [6,7]. HHV-6B reactivation nearly triples the risk of grades II to IV acute graft-versus-host disease (aGVHD), a leading cause of transplant-related death [8]. Additionally, HHV-6B reactivation has been observed during CAR-T-cell production, posing a risk of active viral transmission to immunosuppressed patients [9]. Given the high prevalence of HHV-6B infection and the growing use of transplantation and immunotherapies, the clinical significance of HHV-6B is increasingly apparent, underscoring an urgent need for vaccines and antiviral therapies.

Herpesvirus envelope glycoprotein gB plays a crucial role in mediating viral infection and is a key target for virus neutralization [10,11]. For HHV-6B, entry into susceptible cells—such as T cells, neurons, and salivary gland cells—is mediated by the fusion protein gB and the heterotetrametric gH/gL/gQ1/gQ2 complex. The gH/gL/gQ1/gQ2 complex recognizes the host receptors gp96 and CD134, signaling gB and triggering its conformational change [12,13]. Through a structural transition from a prefusion to a postfusion conformation, gB mediates viral and host membranes fusion, a critical step conserved across herpesviruses [14]. Additionally, HHV-6B gB has been reported to interact with nectin cell adhesion molecule 2 (nectin-2) to

facilitate the infection of salivary gland cells that lack CD134 [15]. As gB not only mediates the essential membrane fusion step but also affects cellular tropism, it represents a promising target for blocking viral infection. Studies on other human herpesviruses have identified monoclonal neutralizing antibodies (nAbs) that target gB and mapped key epitopes, laying a strong foundation for ongoing studies that utilize gB as a candidate antigen in vaccine development [10]. However, for HHV-6B, the lack of a soluble gB expression construct and an unresolved gB structure have significantly impeded understanding of HHV-6B infection mechanisms, hindered nAbs identification, and delayed vaccine development.

In this study, we determined the cryo-electron microscopy (cryo-EM) structure of the HHV-6B gB ectodomain in the postfusion conformation at 2.8 Å resolution. The structure reveals a homo-trimeric complex composed of five distinct domains, and identifies, for the first time, the furin site, which is conserved in multiple herpesviruses. Comparative analysis of sequence and structure identified both conserved and unique features of HHV-6B gB. In addition, we tested cross-binding of nAbs from other herpesviruses and performed a parallel comparison of neutralizing epitopes across other gBs with HHV-6B gB. Although we found limited conservation of these epitopes in HHV-6B gB, we identified regions of potential vulnerability as neutralizing epitopes. Furthermore, immunoprecipitation assay confirmed that nectin-2 interacts with the HHV-6B gB ectodomain, while structural modeling suggests that nectin-2 may also engage prefusion gB. The structural insights into the key fusion protein gB of HHV-6B provide a foundation for developing therapeutic antibodies and vaccines, emphasizing gB's potential vulnerabilities and receptor-binding dynamics.

## Results

### Construct design, purification, and cryo-EM structure determination

We constructed an expression plasmid encoding the ectodomain (residues 1–680) of HHV-6B gB from the Z29 strain, with a C-terminal His-tag for purification. Eukaryotic codon optimization was performed to facilitate expression in 293F cells. Previous studies have shown that hydrophobic fusion loops limit the expression of the herpesvirus gB ectodomain [16–19]. To address this issue, we substituted the hydrophobic residues in the HHV-6B fusion loop regions with nonhydrophobic residues from analogous regions in other gB through multiple sequence alignment (MSA). Specifically, the two fusion loops of HHV-6B gB, $^{102}$VGVV$^{105}$ and $^{187}$WLY$^{189}$, were replaced with $^{102}$HRTT$^{105}$ and $^{187}$ATH$^{189}$, respectively, in the final plasmid construct.

The ectodomain of HHV-6B gB was purified via affinity chromatography, and its purity was assessed via size-exclusion chromatography (SEC) and SDS–PAGE. The SEC profile showed a sharp, single peak, indicating a homogeneous protein output (S1A Fig). SDS–PAGE analysis under nonreducing conditions further confirmed the purity, revealing a single, prominent band above 100 kDa (S1B Fig). The purified HHV-6B gB protein was then subjected to cryo-EM and single-particle analysis to determine its three-dimensional structure. We obtained a high-resolution electron density map and resolved the HHV-6B gB structure at a resolution of 2.8 Å (Figs 1A and S2 and S1 Table).

### Overall structure of HHV-6B gB

Like other herpesvirus gB, HHV-6B gB forms a typical trimer with three protomers tightly packed together, resembling a long, rod-like structure. The trimer has a height of 162 Å (Figs 1B and S3A). As determined from the top view, the three subunits separate and protrude outward, creating a central cavity with long and short axes measuring 41 Å and 21 Å, respectively. In contrast, the bottom view revealed that the subunits interlock with each other.

The ectodomain of each HHV-6B gB protomer is divided into five distinct structural domains (Fig 1C). From the N-terminus, the sequence starts with Domain IV (DIV), positioned at the top. It then extends downward through Domain III (DIII) and Domain II (DII), reaching Domain I (DI), which is located near the membrane. After passing through DI, the structure folds back upward, traversing DII and DIII again before returning to DIV at the top. DIV is then connected to Domain V (DV), which runs parallel to DIII along the vertical axis (Fig 1D). Each protomer contains five intrachain disulfide bonds, with no interchain disulfide bonds present in the trimer. The furin site, a common feature in multiple herpesvirus

PLOS Pathogens

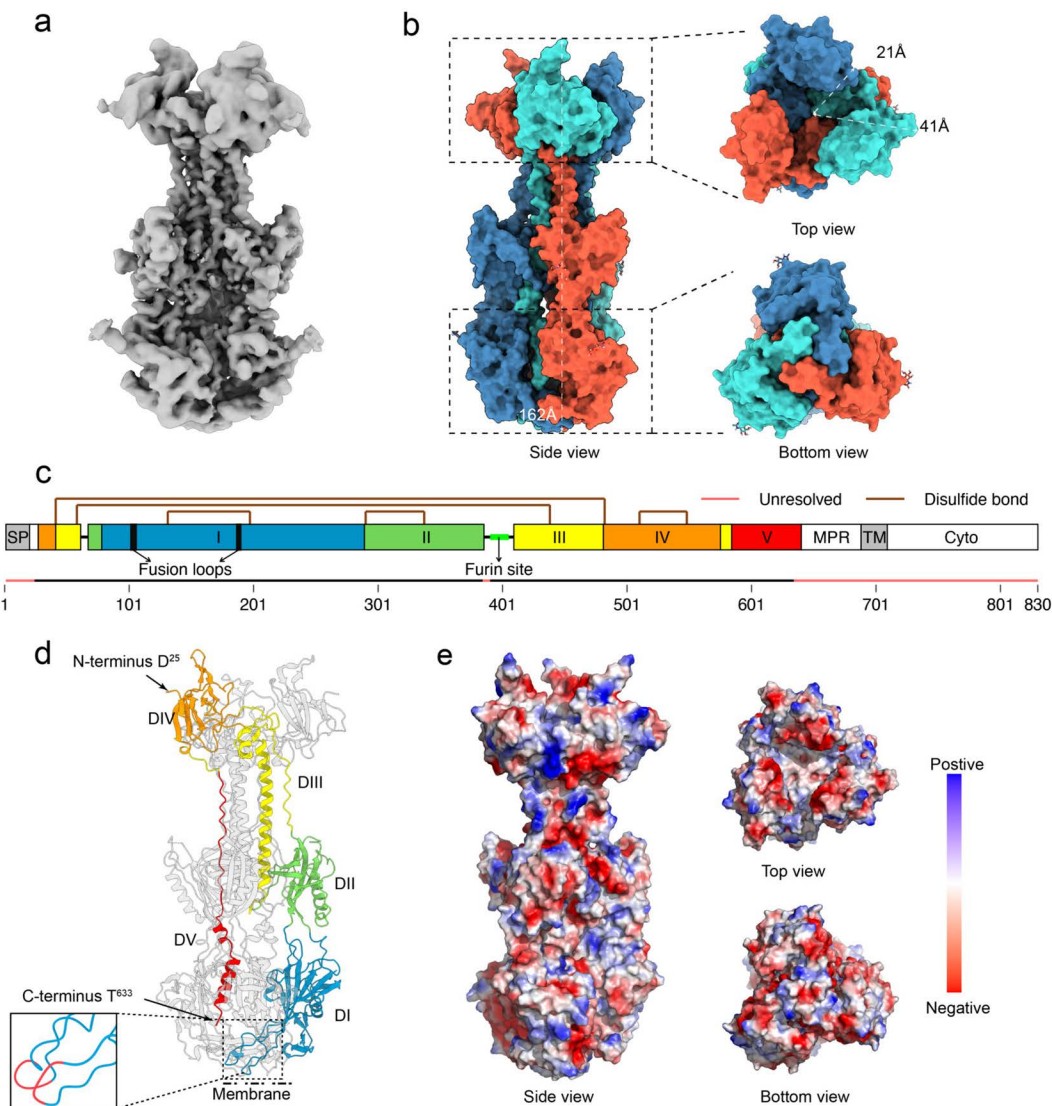

**Fig 1. Structure of the HHV-6B gB ectodomain in the postfusion conformation. a**, Cryo-EM density map of HHV-6B gB. The 3D density map at 2.8 Å resolution reveals the postfusion trimeric structure of HHV-6B gB. **b**, Structural overview of the HHV-6B gB trimer. The structure of the HHV-6B gB trimer is presented in three orientations. Each protomer is shown in a different color, and key distances between structural elements were measured via ChimeraX v1.6 and are labeled in the figure. **c**, Full-length HHV-6B gB sequence and domain layout. The sequence of HHV-6B gB is segmented into distinct structural domains, highlighted in unique colors. Key features, including the fusion loops, furin cleavage site, regions not resolved in the structure, and intrachain disulfide bonds, are marked. SP, signal peptide; MPR, membrane proximal region; TM, transmembrane region; Cyto, cytoplasmic tail. **d**, Cartoon representation of the HHV-6B gB trimer. The trimeric structure of HHV-6B gB is depicted, with each domain colored as in panel (c). The N- and C-termini, the positions of the fusion loops, and the likely orientation relative to the viral membrane are indicated. **e**, Electrostatic potential map of HHV-6B gB. The electrostatic surface potential of the gB trimer is shown from various perspectives, with regions colored to indicate negative (red), neutral (white), and positive (blue) charges, as per the scale bar.

gB, typically harbors the consensus sequence R-X-K/R-R, where X represents any amino acid. In HHV-6B gB, the furin site is located at amino acids 399/400 and is characterized by the sequence $^{396}$RRRR|D$^{400}$. The electrostatic potential map of gB does not reveal significant clusters of positive or negative charges (Fig 1D).

DI, spanning residues I⁷⁹–V²⁸⁹, forms the primary structural base of the extracellular region of gB (S3B Fig). In wild-type gB, this domain contains two highly hydrophobic fusion loops critical for mediating membrane fusion. In our construct, these two fusion loops, located at the bottom of DI, were mutated. DII, composed of two discontinuous segments, A⁶⁵–N⁷⁸ and C²⁹⁰–V⁴⁰⁴, is positioned above DI and on the outer side of the base of DIII. It is connected to DIII via two long, unstructured linkers, with the furin site ³⁹⁶RRRR|D⁴⁰⁰ located within one of the linkers (S3B Fig). DIII consists of three segments: S⁴²–N⁶⁴, K⁴⁰⁵–K⁴⁸¹, and I⁵⁷⁹–A⁵⁸⁴. The DIII long α-helix of each protomer together establishes a central coiled coil of the trimeric gB, and at its apex, DIII folds back nearly 180°, which contributes to a six-helix arrangement at the apex. DIV, comprising the N-terminal segment D²⁵–C⁴¹ and residues C⁴⁸²–D⁵⁷⁸, forms an ear-like protrusion at the top of this triangular arrangement, constituting the crown of the entire gB structure. DV, spanning residues F⁵⁸⁵–T⁶³³, is characterized by a loosely folded loop and is positioned internally within the gB trimer. It extends downward along the same axis as DIII and forms two short α-helices near its terminus. The lower part of DV is embedded within the clefts between DI protomers, further stabilizing the base of the gB. The regions encompassing residues 20–24, 384–390, and 632–680 were not resolved in the structure, indicating potential flexibility or disorder in these segments.

The structure of HHV-6B gB reveals a conserved overall architecture that is consistent with that of other herpesvirus gB in postfusion conformation. We further compared our experimentally determined structure with an AlphaFold3-predicted model of HHV-6B gB. The predicted model aligned well with the cryo-EM structure, with a root-mean-square deviation (RMSD) of 1.386 Å, capturing the domain organization and overall fold (S3A Fig). Notable differences were observed in the linker between DII and DIII containing the furin cleavage site, which exhibited a shifted position compared to the experimental structure, and in the DI fusion loops, which adopted different orientations between the predicted and experimental models (S3A Fig).

We next explored the unique features of the HHV-6B gB structure that distinguish it from those of other herpesvirus gB.

## Structural insights into the N-terminal region and furin site

In our analysis of the HHV-6B gB structure, we were able to observe its N-terminal region starting from residue D²⁵, which is distinguished from other herpesvirus gB structures (Fig 2A). Typically, the structures of gB from other herpesviruses are discernible from residues further from the N-terminus, such as those of herpes simplex virus 1 (HSV-1) and varicella-zoster virus (VZV), gB, which are discernible from residues 111 (PDB: 2GUM) and 115 (PDB: 6VLK), respectively, as their N-terminal regions may fail to form stable, crystallizable structures [20,21].

Sequence alignment revealed significant variability in the N-terminal sequences of herpesvirus gB, in both length and amino acid composition, and the first β-strand at the N-terminus within the DIV is conserved across various herpesviruses (Figs 2A and S4). This β-strand is followed by a cysteine residue that forms a disulfide bond with a downstream cysteine, although the sequence position and amino acids forming this strand can vary. Among human herpesviruses with resolved gB structures, HHV-6B gB has the shortest peptide segment preceding the first β-strand, followed by Epstein-Barr virus (EBV), Kaposi's sarcoma-associated herpesvirus (KSHV), and human cytomegalovirus (HCMV), whereas HSV-1 and VZV have the longest segments. In our resolved structure, the HHV-6B gB N-terminus, starting shortly after the predicted signal peptide (residues 1–19), directly forms a short β-strand with residues ²⁷YIR²⁹ that contributes to the formation of DIV. The corresponding first β-strands in other herpesvirus gB include ¹¹³FYV¹¹⁵ in HSV-1, ¹²⁰YV¹²¹ in VZV, ⁹¹RVCS⁹⁵ in HCMV, ⁴⁸FR⁴⁹ in EBV, and ⁶⁶RV⁶⁷ in KSHV (Fig 2A). The early involvement of the HHV-6B gB N-terminus in DIV formation explains why we could observe more of N-terminal region in this structure than other gB. In addition, although the DIV is composed largely of β-strands across all herpesvirus gB, the spatial arrangement of the first β-strand in HHV-6B gB DIV differs from other herpesvirus (Fig 2A). In HHV-6B, the first β-strand is positioned on the outer edge of the DIV and interacts with the central region of DIV to create an antiparallel β-sheet. In contrast, the first β-strands of other herpesvirus gB are positioned near the center of the top of DIII, close to the junction between the two DIII central helices, and are often flanked by two oppositely oriented β-strands. The unique sequence and structural features of the N-terminus of HHV-6B gB may contribute to its distinct functional and immunogenic properties.

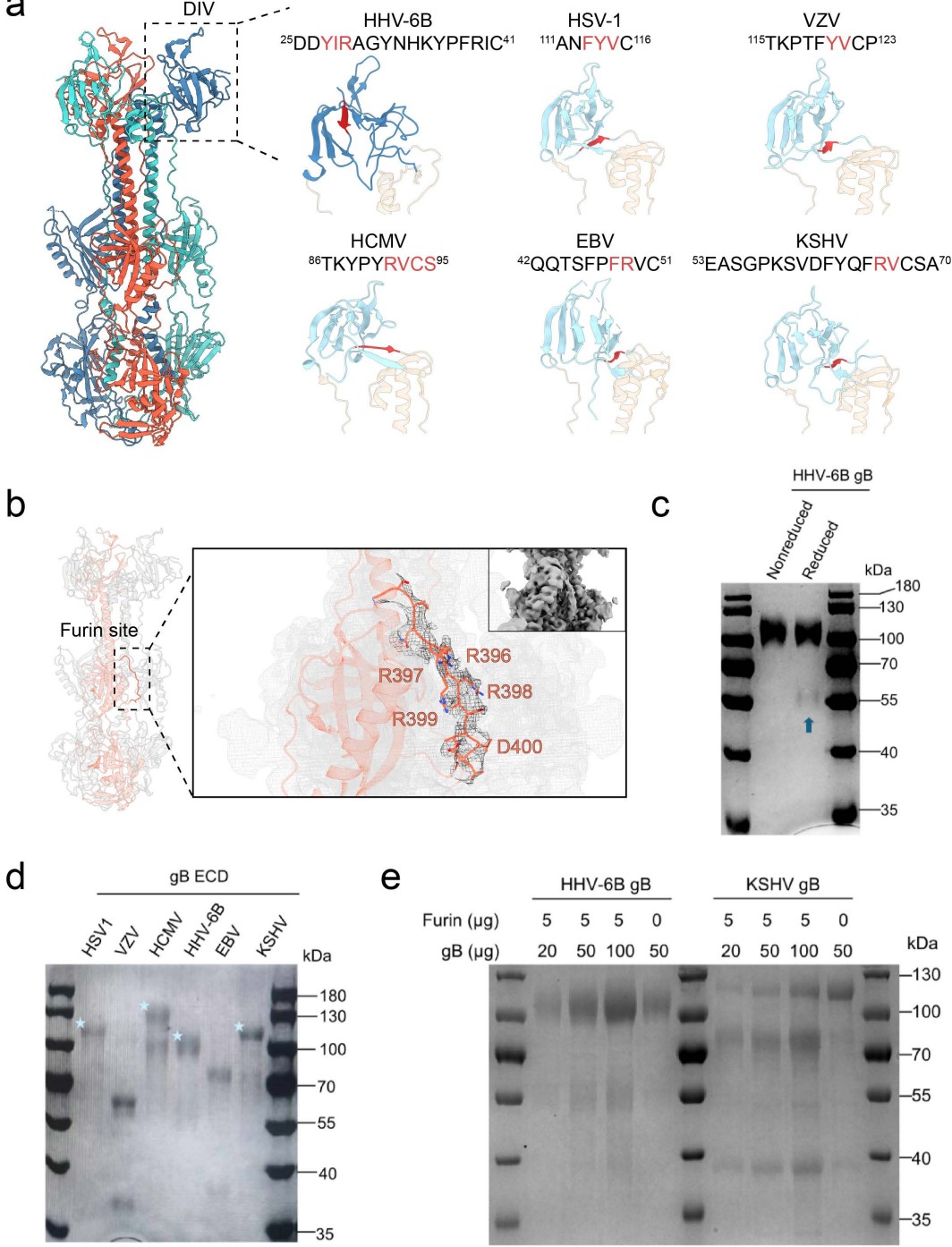

**Fig 2. Structural and furin cleavage properties of HHV-6B gB and its comparison with other herpesvirus gB. a**, Sequence and structural comparison of N-terminal regions in gB DIVs. The structures of DIV and part of DIII from HHV-6B gB, along with other herpesvirus gB [HSV-1 (PDB: 2GUM), VZV (PDB: 6VLK), HCMV (PDB: 5CXF), EBV (PDB: 3FVC), and KSHV (PDB: 8Y48)], are displayed [17,19–22]. The corresponding N-terminal sequences of the DIVs are shown, starting from the first resolvable amino acid in each structure. The first β-strand of each N-terminal region is highlighted in red in both the structure and sequence. **b**, Structural details of the furin site in HHV-6B gB. A magnified view of one protomer's furin site, the residues 396RRRR|D400 in the HHV-6B gB trimer, is shown, with the corresponding electron density map overlaid to provide structural detail of this region. **c**, Coomassie-stained SDS-PAGE analysis of HHV-6B gB under reducing and non-reducing conditions. Blue arrow indicates the cleaved portions of

the reduced HHV-6B gB sample. **d**, SDS-PAGE analysis of reduced ectodomain (ECD) preparations of purified gB from various herpesviruses without mutations introduced at their respective furin sites. Light blue asterisks indicate uncleaved portions of each gB. **e**, *In vitro* furin cleavage assay of purified HHV-6B and KSHV gB ectodomains. gB-His proteins were incubated with or without purified furin, and cleavage products were analyzed by SDS-PAGE under reducing conditions.

We also successfully resolved the structure of the furin site in HHV-6B gB (Fig 2B). Furin cleavage of gB reportedly affects receptor recognition, gB folding, cellular infection, and *in vivo* pathogenesis differently across various herpesviruses [23–26]. However, this region has not been resolved before in gB structural studies, including those with design to avoid furin cleavage [17,19,21,22,27,28]. This could be ascribed to the inherent flexibility, which complicates structural determination. The absence of structural data for this region has limited our understanding of how the furin site contributes to the functional role of gB.

For our HHV-6B gB construct, which did not include mutations to prevent furin cleavage, we observed a relatively low proportion of cleaved protein, and the structure we resolved corresponds to the intact furin site (Fig 2B and 2C). The electron density map reveals a relatively flexible but discernible region corresponding to this loop, which was identified as the furin site. Our structural analysis of the HHV-6B gB $^{396}$RRRR|D$^{400}$ furin site is consistent with previous hypotheses, showing that it is an exposed loop on the exterior of the gB trimer that does not interact with the main gB body [29]. Although the structure of residues 384–390, located before the furin site, was not resolved, our observations still illustrate how DII connects to DIII through a long linker that includes the furin site.

Although HHV-6B gB contains a furin consensus sequence, cleavage may still be dependent on the surrounding sequences and the specific conditions of protein expression. To explore this further, we applied a predictive model that assesses furin cleavage potential on the basis of both recognition motifs and the surrounding sequence context to estimate furin cleavage potential across various herpesvirus gB, which could reflect the likelihood of furin-mediated cleavage at specific arginine or lysine sites [30]. As expected, except HSV-1 gB, other herpesvirus gB are predicted to have cleavage sites within the linker region between domains DII and DIII (S5A Fig). Notably, HHV-6B gB exhibited the lowest cleavage potential among the gB predicted to have furin sites, suggesting that the segment containing the furin consensus sequence, residues $^{393}$VNLRRRR|DL$^{401}$ of HHV-6B gB, may be less favorable for furin recognition and cleavage.

Building on this prediction, we further compared the cleavage status of purified extracellular gB across these viruses. We found that HSV-1 gB, lacking a furin site, was not cleaved, whereas other gB were partially or fully cleaved, with HHV-6B and KSHV gB exhibiting the lowest cleavage (Fig 2D). Furthermore, *in vitro* furin cleavage assays demonstrated that HHV-6B gB is more resistant to cleavage than KSHV gB. After 3 hours of digestion at 37°C with purified furin, KSHV gB was predominantly cleaved, whereas HHV-6B gB remained largely intact (Fig 2E). To further evaluate whether the low cleavage efficiency of HHV-6B gB is attributable to its furin site sequence, we replaced the native cleavage motif with the corresponding sequence from EBV or VZV and analyzed the cleavage status of the resulting chimeric proteins after expression and purification. Both substitutions led to increased levels of cleaved protein, supporting that the HHV-6B furin site sequence could be suboptimal for proteolytic processing (S5B Fig).

Our findings show that the furin site in HHV-6B gB is structurally intact and relatively resistant to cleavage, which is distinct from other herpesviruses. Given the conservation of the furin site and its established role in multiple herpesviruses, the presence of a furin site in HHV-6B gB indicates a potentially important but currently undefined function. Interestingly, the segment containing an intact furin site, rich in polar amino acids and exposed on the surface of the gB trimer, suggests that it may interact with host membrane proteins, potentially influencing viral entry or other aspects of the infection cycle. Moreover, as a small proportion of HHV-6B gB can still be cleaved by furin (Fig 2C), it remains to be determined whether the cleaved and uncleaved forms of gB exhibit different functions. These observations highlight the need for further investigation into the broader role of the HHV-6B gB furin site in modulating virus–host interactions and membrane fusion mechanisms.

## Comparative analysis of gB conservation and variability across HHV-6B and other herpesviruses

Given the high degree of gB conservation between HHV-6A and HHV-6B, with over 90% sequence identity, our structural analysis of HHV-6B gB also offers valuable insights into the structure and function of HHV-6A gB. This high sequence conservation suggests that functional regions are likely shared between the two viruses, but it also raises questions about the structural distribution of the minor sequence differences, which could affect the infection dynamics of HHV-6A and HHV-6B.

To delineate the structural implications of these sequence variations, we analyzed the gB sequences spanning residues 20–680, incorporating sequences from multiple HHV-6B strains as well as HHV-6A strains. The sequence variations were categorized into three groups and mapped onto the resolved structure of the HHV-6B Z29 strain gB to visualize their distribution. These groups include the following: (1) conserved across all analyzed strains, (2) conserved within HHV-6B strains, and (3) variable among HHV-6B strains (Fig 3A and S2 Table).

In total, 564 of the 600 resolved residues and 59 of the 61 unresolved residues in the analyzed segment are globally conserved across all included strains of both HHV-6A gB and HHV-6B gB, indicating that approximately 94% of the residues fall into the first group. The second group of residues is primarily concentrated in DI and DII, with an additional site at residue 600 in DV. They constitute a significant part of the interspecies variability between HHV-6A and HHV-6B and may contribute to species-specific receptor interactions or immune evasion mechanisms. The third group, containing residues that vary among HHV-6B strains, is predominantly found in DIV and parts of DII and DI. These variable residues may reflect adaptations to different host environments or selective pressures within species. While the functional consequences of the sequence variations in DI, DII, and DIV, which are relatively exposed in the gB trimer, are not fully understood, previous studies have shown their impact on antibody binding and neutralization. For example, residue 347 on the outer long α-helix of DII, which is consistently N$^{347}$ in HHV-6A but changes to K$^{347}$ or T$^{347}$ in HHV-6B, was found to determine the specificity of the monoclonal antibody 87-y-13, which neutralizes HHV-6A but not HHV-6B [31,32]. This finding suggests an important role of exposed-region sequence variations in antibody recognition and viral immune evasion. In contrast, DIII and DV form the structural core of gB, with DIII contributing to the central α-helix and DV primarily comprising the extended coil in the postfusion gB structure, and they are highly conserved between HHV-6A and HHV-6B and across their strains. This high degree of conservation indicates that DIII and DV likely play a shared critical role in maintaining the structural and functional integrity of gB in both viruses.

Building on our analysis of HHV-6A and HHV-6B gB, we extended the comparison to other herpesvirus gB to investigate distinct or shared features. We compared the sequence and structure of HHV-6B gB with those of other herpesvirus gB, including pseudorabies virus (PrV), HSV-1, VZV, HCMV, EBV and KSHV, calculated the conservation score of amino acids, and mapped the conservation score to the resolved structure of HHV-6B gB (Fig 3B). Although there is significant sequence variation among different gB, particularly within the segment from the N-terminal signal peptide to the beginning of DIV, the DII to DIII linker that contains the furin site and the C-terminal intracellular segment, 71 amino acids of the full-length HHV-6B gB (71/830) are conserved across all analyzed herpesvirus gBs (S4 Fig). Among them, 10 cysteines that form intrachain disulfide bonds are shared among the seven herpesviruses, underscoring the importance of these disulfide bonds in gB folding, conformational changes, and functions (Fig 3B). We observed that the highly conserved residues across different herpesviruses are often located internally within each domain and the core of the gB trimer (Figs 3B and S6A).

Structurally, HHV-6B gB exhibits the lowest RMSD value compared with EBV gB, followed by HCMV gB, KSHV gB, and the gB of the α-herpesviruses PrV, HSV-1 and VZV. These findings suggest that the gB of HHV-6B, a β-herpesvirus, also shares features of γ-herpesviruses, especially EBV (Figs 3C and S6B). The segment of HHV-6B gB from the N-terminal signal peptide to the start of DIV is significantly shorter than those of HCMV, PrV, VZV, and HSV-1 gB but relatively closer in length to EBV and KSHV gB (S4 Fig). When the five domains are superimposed, their structures align well, indicating conserved folding, although differences in angles and the number of amino acids involved in specific folds exist (Fig 3C).

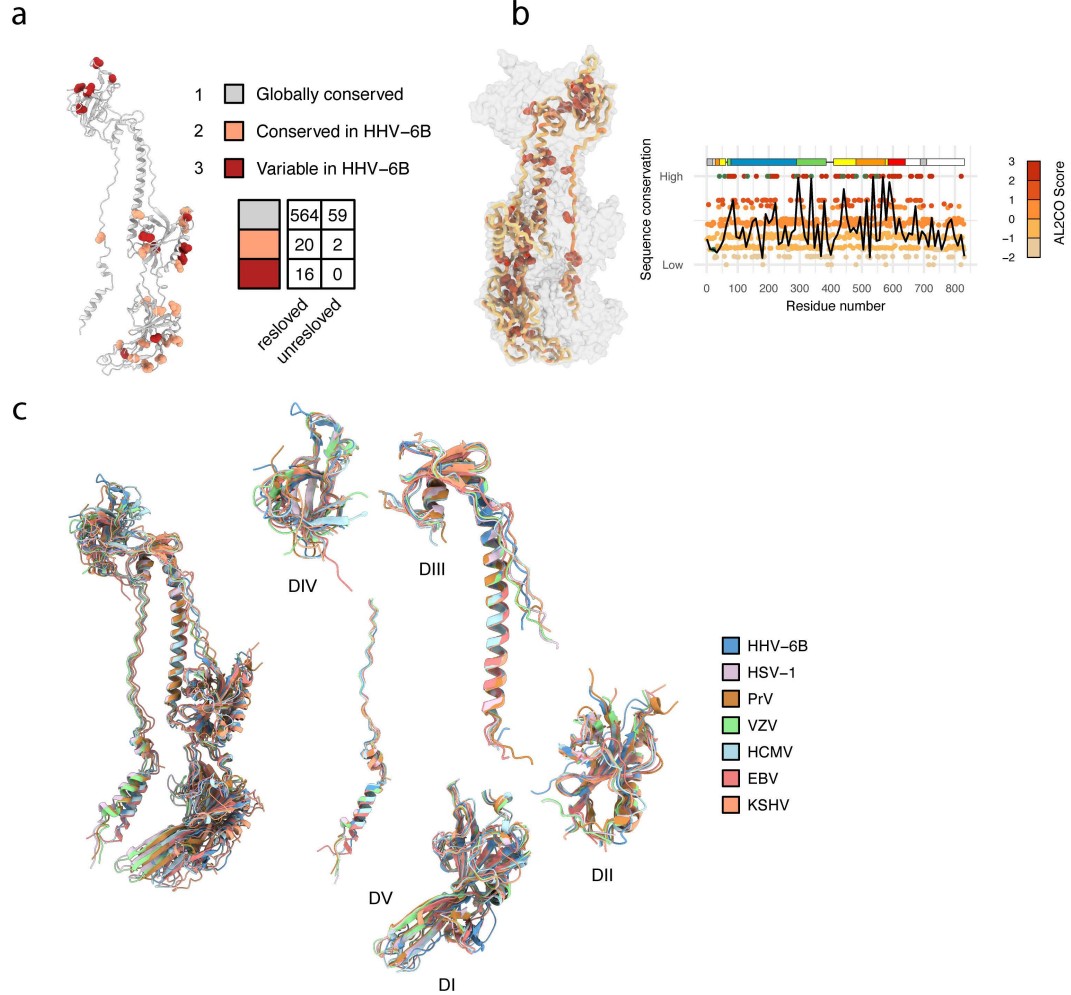

**Fig 3. Intra- and interspecies sequence and structural conservation analysis of HHV-6B gB. a**, Sequence comparison between HHV-6A and HHV-6B gB mapped onto the HHV-6B gB structure. Multiple sequence alignment of the gB ectodomain (residues 20-680) from HHV-6A and HHV-6B strains was performed. The aligned amino acids were categorized into three groups: (1) conserved across all analyzed strains, (2) conserved within HHV-6B strains, and (3) variable among HHV-6B strains. These categories are distinguished by different colors as indicated, and mapped onto the resolved HHV-6B gB structure, with residues in the second and third groups represented as spheres to indicate their positions on the structure. The table on the right summarizes the number of residues in each group, distinguishing between resolved and unresolved regions within HHV-6B gB residues 20-680. **b**, The gB sequence of HHV-6B gB was aligned with those of PrV, HSV-1, VZV, HCMV, EBV, and KSHV, and conservation scores were calculated to generate a conservation map. In the conservation map, each circle represents an amino acid at the corresponding position in the HHV-6B gB sequence, with cysteine residues highlighted by green outlines. Above the map, a domain segmentation bar is provided to indicate the structural domains corresponding to the position of the sequence, using the same color scheme as in Fig 1B. On the right, the conservation scores are mapped onto the structure of an HHV-6B gB protomer displayed as a cartoon, visualizing how the sequence conservation is related to its structure. **c**, Structural comparison of gB from herpesviruses. The HHV-6B gB structure, as resolved in this study, was compared to the structures of gB from five other herpesviruses: PrV (PDB: 6ESC), HSV-1 (PDB: 2GUM), VZV (PDB: 6VLK), HCMV (PDB: 5CXF), EBV (PDB: 3FVC), and KSHV (PDB: 8Y48) [17,19–22,27].

For example, while DV generally exhibits a loose extended structure with two α-helices forming a defined angle, both the length and the angle of these two α-helices vary among herpesviruses (Fig 3C). Excluding KSHV gB, in which this region is not fully resolved, the number of amino acids forming the short α-helix ranges from 6 to 9, whereas the long α-helix ranges from 14 to 19 across the other five herpesviruses (S6C Fig). For HHV-6B gB, the short and long α-helices in DV are composed of 7 and 14 amino acids, respectively. The angle between these two helices also varies across different

herpesvirus gB, ranging from around 55° to 66° (S6C Fig). When the overall ectodomains of the gB are superimposed, some structural misalignments are observed, primarily due to differences in the orientations between the domains within the trimer, which highlights the structural divergences between different herpesviruses (Fig 3C).

Our analysis of gB sequence conservation between HHV-6A and HHV-6B, coupled with the resolved HHV-6B gB structure, highlights the marked conservation of DIII and DV, whereas variations in DI, DII, and DIV account for the majority of interspecies and intraspecies differences. This pattern aligns with previous findings in HCMV gB and indicates that DIII and DV, which form the core of gB, may be conserved and essential for maintaining structural integrity, whereas the more exposed DI, DII, and DIV are likely involved in adaptive interactions with host receptors and immune evasion mechanisms. Interestingly, this is consistent with our observation that conserved residues across different herpesviruses tend to cluster internally within the gB structure. Additionally, our comparative analysis of HHV-6B gB with other herpesvirus gB revealed that while HHV-6B gB shares typical herpesvirus gB features, its structural characteristics, such as domain length and angles, could vary, and HHV-6B gB shows a unique blend of β- and γ-herpesvirus traits.

## gB neutralizing antibody epitope comparison with HHV-6B gB

gB is recognized as a critical target for nAbs against herpesviruses. However, no HHV-6B gB-specific nAbs have been reported, and the neutralizing epitopes of HHV-6B gB remain poorly understood. In other herpesviruses, several nAbs have been structurally characterized in complex with gB, typically binding to DI, DII, and DIV. Given the role of gB as the conserved core infection machinery across herpesviruses, as well as the conserved nature of its sequence and structure, certain structural elements in gB may serve as potential cross-neutralization targets. Additionally, the known neutralizing epitopes in other herpesvirus gB could implicate vulnerability of HHV-6B gB and inform the development of HHV-6B gB nAbs. Here, we investigated whether existing nAbs from other herpesviruses exhibit cross-reactivity with HHV-6B gB and aimed to identify potential neutralizing epitopes on HHV-6B by comparing known neutralizing epitopes from other herpesvirus gB with the corresponding regions on HHV-6B gB.

The antibodies analyzed included HDIT102, which binds to DI of HSV-1 gB; 93K, which targets DII of VZV gB; 1G2 and SM5–1, which bind to DI and DII of HCMV gB, respectively; and 3A3 and 3A5, which target DII and DIV of EBV gB (Fig 4A and 4B) [33–37]. These nAbs and the ectodomains of their targeted gB were expressed and purified for a biolayer interferometry (BLI) assay to assess the binding of these nAbs to their original targets and to HHV-6B gB. No cross-reactivity of these nAbs with HHV-6B gB was observed (S7 Fig). To explore this further, we displayed the binding footprints of these nAbs on their respective gB and compared these epitopes with aligned regions on the resolved HHV-6B gB structure, followed by an analysis of key binding residue conservation and domain structural similarity. Additionally, resolved N-linked glycosylation sites in HHV-6B gB and the reported nAb-gB complex structures were also annotated to assess their potential impact on epitope accessibility. Although the overall structures of the analyzed domains of different herpesvirus gB and HHV-6B gB are highly similar, especially for DII and DIV, we observed significant differences between the sequences corresponding to these antibody epitopes on HHV-6B gB and the original gB sequences.

Specifically, the aligned footprints of HDIT102 and 1G2 partially overlap on HHV-6B gB (Fig 4C). The HDIT102 footprint on HHV-6B gB, which was originally mapped from HSV-1 gB, covers mainly parts of the outer α-helix, the β-strands above the helix in DI, and a small portion of the loop region above the β-strand. In contrast, the 1G2 footprint primarily occupies the loop region adjacent to HDIT102's footprint, including a small portion of the β-strand below the loop. The two footprints overlap at residues $G^{229}$, $S^{230}$ and $I^{244}$ on HHV-6B gB, indicating the functional importance of this region. In comparison, the HDIT102 footprint covers a relatively large area but shows limited conservation between HSV-1 and HHV-6B gB (Fig 4B). Besides, we observed that $N^{247}$ in HHV-6B gB, aligned with $D^{323}$ that contact HDIT102 in HSV-1 gB, is glycosylated in our structure, which may further reduce accessibility of this epitope. Interestingly, two residues of the overlapping aligned footprints of the two nAbs, $G^{229}$ and $I^{244}$, are conserved between HCMV and HHV-6B gB (Fig 4C). In HCMV, the corresponding residue $G^{282}$ directly interacts with the heavy chain of 1G2, suggesting the potential of this site

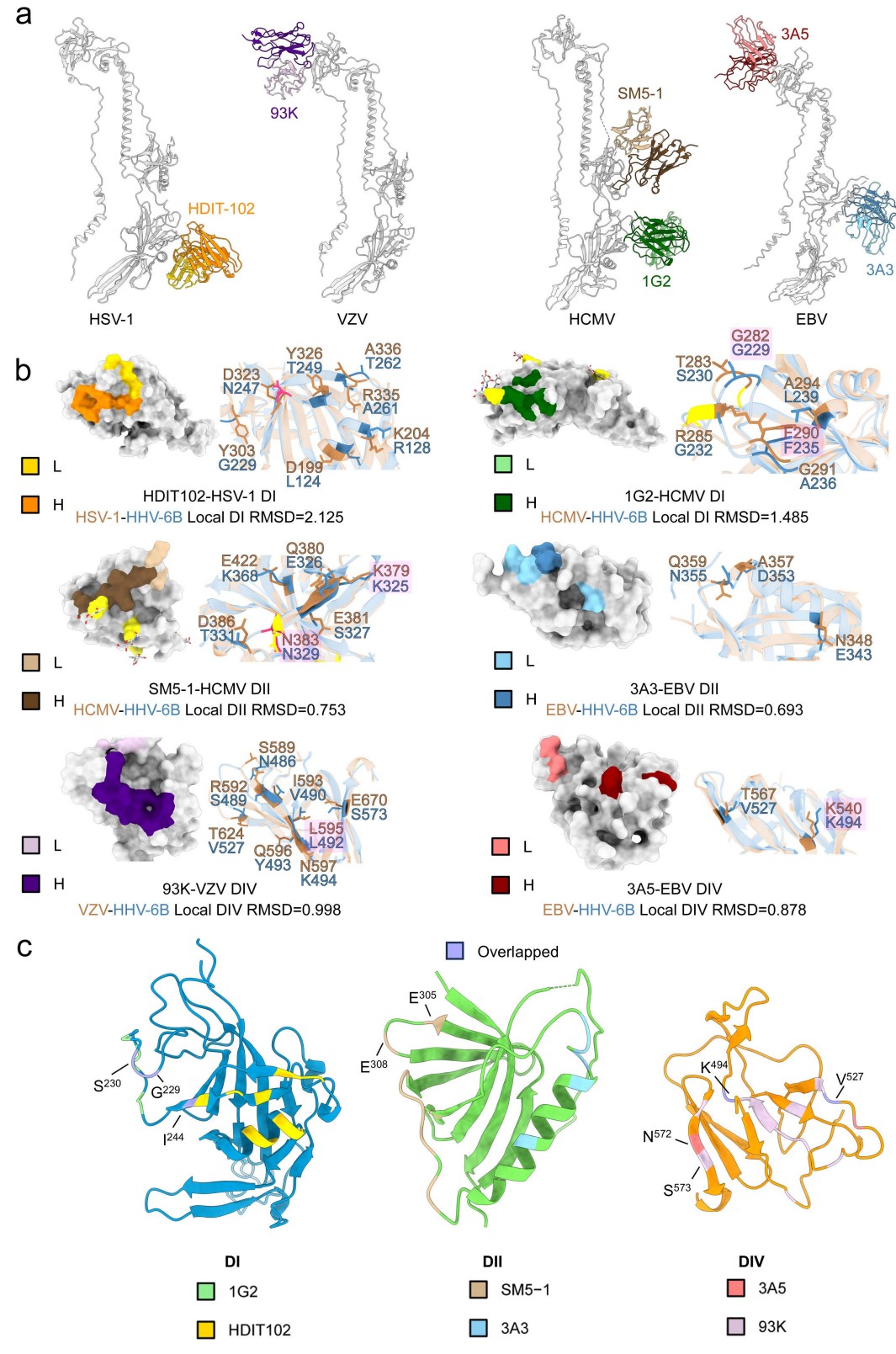

**Fig 4. Neutralizing antibody epitope analysis of HHV-6B gB. a,** Herpesvirus-neutralizing antibodies binding to their respective gB. The figure includes HDIT102 bound to HSV-1 gB DI (PDB: 8RH0), 93K bound to VZV gB DIV (PDB: 6VN1), 1G2 bound to HCMV gB DI (PDB: 5C6T), SM5-1 bound to HCMV gB DII (PDB: 7KDD), and 3A3 and 3A5 bound to EBV gB DII and DIV (PDB: 7FBI), respectively. The light and heavy chains of each antibody are color-coded as in Fig 4B. **b,** Local comparison of neutralizing epitopes from other herpesvirus gB with aligned HHV-6B gB regions. The footprints of the neutralizing antibodies from Fig 4A on their respective herpesvirus gB domains are displayed and colored as indicated. The corresponding HHV-6B gB domain is aligned with the original gB domain from other herpesviruses, and the RMSD values (in Å) for the alignments are provided. The aligned interaction regions are shown in cartoon format, with HHV-6B gB depicted in blue and the original gB structures in light brown. Key interacting residues from the original gB and their aligned counterparts in HHV-6B gB are labeled, with conserved residues highlighted. Resolved glycans within the displayed regions are shown as sticks in the surface views. Glycosylated residues identified in the antibody–gB complex structures are colored bright yellow, while glycosylated residues resolved in the HHV-6B gB structure are colored hot pink, both in the surface and cartoon representations. **c,** Footprints of neutralizing antibodies aligned to corresponding regions of HHV-6B gB. HDIT102 and 1G2 footprints were aligned to DI of HHV-6B gB, SM5-1 and 3A3 to DII, and 93K and 3A5 to DIV, as shown in the indicated colors. Overlapping or adjacent residues resulting from aligned antibody footprints on the same HHV-6B domain are labeled.

as a cross-β-herpesvirus-neutralizing target. The RMSD between DI of HHV-6B and HCMV gB is also smaller than that between HHV-6B and HSV-1 gB, measuring 1.485 Å and 2.125 Å, respectively, indicating greater structural similarity between HCMV gB DI and HHV-6B gB DI (Fig 4B). Although DI is structurally similar between HHV-6B gB and HSV-1 gB or HCMV gB, the folding of these footprint regions shows slight angular deviations in different gB.

For antibodies targeting gB DII, the footprints of SM5–1 and 3A3 were aligned to different regions on HHV-6B gB DII (Fig 4B and 4C). SM5–1's footprint is located on the linker region preceding the DII long α-helix and nearby β-strands on the outer side of DII, whereas 3A3's footprint covers the long α-helix and the linker extending above the helix. These two footprints do not overlap and reside on different surfaces of DII. For SM5–1, the footprints of both its light and heavy chains have residues conserved between HCMV and HHV-6B. The light-chain footprint is smaller and completely conserved between the two viruses, containing residues $E^{359}$ and $E^{361}$ in HCMV gB, which correspond to $E^{305}$ and $E^{308}$ in HHV-6B gB (Fig 4C). Additionally, residues $K^{379}$ and $N^{383}$ in HCMV gB, which are involved in direct binding to SM5–1, are also conserved. Of note, $N^{383}$ in HCMV gB as a contact residue for SM5–1, is glycosylated in the resolved structure, suggesting that glycosylation and antibody binding may not be mutually exclusive and may depend on local glycan orientation. The aligned $N^{329}$ in our HHV-6B gB structure is also glycosylated. However, among the six amino acids that directly interact with SM5–1, only two are conserved between the two viruses, possibly explaining why SM5–1 did not cross-interact with HHV-6B gB. For 3A3, no directly binding residues were conserved on HHV-6B gB. Overall, the structural similarity of DII between HHV-6B and HCMV or EBV is high, with RMSD values of 0.753 Å and 0.693 Å, respectively (Fig 4B).

For 93K and 3A5, which bind to DIV of gB, the aligned footprints on HHV-6B gB are partially close or overlapping, with both heavy-chain footprints containing the same β-strand, where residues $S^{573}$ (93K) and $N^{572}$ (3A5) are adjacent (Fig 4C). The aligned footprints of both nAbs also share the β-strand linker regions at residues $K^{494}$ and $V^{527}$. DIV forms the crown structure at the top of the trimer, consisting of a short segment following the N-terminal disordered region and a longer segment from the intermediate to distal ectodomain. Apart from a small binding region of 93K located in the linker region following the first β-strand of the N-terminus, the footprints of 93K and 3A5 are distributed across other β-strands and linkers formed by the later longer segment, with their corresponding residues on HHV-6B DIV having two or one conserved amino acid compared with VZV and EBV gB DIV (Fig 4B). The RMSD values were 0.998 Å and 0.878 Å between DIV of HHV-6B gB and that of VZV and EBV gB, respectively.

Overall, despite a certain number of shared residues and degree of domain structural similarity between HHV-6B and other herpesvirus gB, the sequence divergence in the epitopes limits the cross-reactivity of existing nAbs targeting other herpesvirus gB with HHV-6B gB. Notably, we identified specific regions where epitopes from different herpesvirus gB nAbs aligned to the same or adjacent sites on HHV-6B gB, which may be potentially vulnerable sites that could serve

as promising targets for screening HHV-6B gB nAbs. Interestingly, none of the analyzed antibody footprints aligned with highly conserved residues across herpesviruses. Whether these universally conserved residues among herpesviruses hold value as broad-spectrum neutralization targets remains to be studied.

### Identification and structural modeling of the interaction of HHV-6B gB with the nectin-2 receptor

In addition to its role as a fusion protein that mediates virus–cell membrane fusion via conformational changes, herpesvirus gB has been reported to engage with various cellular receptors to facilitate the infection process. HSV-1 gB has been reported to bind non-muscle myosin heavy chain IIA (NMHC-IIA), non-muscle myosin heavy chain IIB (NMHC-IIB), paired immunoglobulin-like type 2 receptor alpha (PILRα), and myelin-associated glycoprotein (MAG), whereas gB of EBV and KSHV bind to neuropilin 1 (NRP1) [38–44]. HHV-6B gB has been reported to interact with nectin-2, which can serve as an alternative receptor in cells lacking the gH/gL/gQ1/gQ2 tetramer receptor CD134, thus mediating infection [15]. However, the structural details of herpesvirus gB-receptor complexes remain unclear.

As the V-set domain of nectin-2 reportedly interacts with HHV-6B gB, we first expressed and purified the V-set protein with an Fc tag (V-set-Fc) and confirmed its interaction with HHV-6B gB via an immunoprecipitation assay (Figs 5A, S8A and S8B). Building on this, we employed the reported structure of the nectin-2 V-set domain (PDB: 3R0N) and our resolved postfusion structure of the HHV-6B gB ectodomain to predict their interaction via AlphaFold2 [45]. The V-set domain, which is composed predominantly of β-strands, forms hydrogen bonds through residues $R^{35}$ and $N^{137}$ on two of its β-strands, with $E^{150}$, $N^{152}$ and $Y^{132}$ located in DI of HHV-6B gB in the model with the highest predictive score (Fig 5B). A combined mutations of these predicted interacting residues in HHV-6B gB partially reduced its interaction with V-set in immunoprecipitation assay, supporting this predicted model (Fig 5C). This interaction region on HHV-6B gB DI is positioned on the lower lateral side of the domain, with these three residues being fully conserved across all analyzed strains of HHV-6A and HHV-6B. However, neighboring residues at positions 148, 156, and 159 are conserved within the strains of each virus but differ between HHV-6A and HHV-6B, raising questions about the role of nectin-2 as an HHV-6A gB receptor.

Interestingly, in the resolved prefusion structure of HCMV gB, DI is similarly exposed and retains its overall domain integrity, a feature corroborated by the binding of neutralizing antibodies that target DI in various herpesvirus gB. This led us to explore whether the V-set might also bind to the prefusion conformation of HHV-6B gB. With the resolved prefusion structure of HCMV gB (PDB: 7KDP) as a template, we constructed a prefusion model of HHV-6B gB via SWISS-MODEL (Fig 5D) [46]. Structural comparison of DI between the prefusion and postfusion conformations revealed significant similarity (RMSD = 1.720 Å), with significant conformational changes occurring at the fusion loop region near the base of DI. Notably, the predicted V-set binding site on the external lateral side remains largely similar and accessible in the prefusion conformation, suggesting the potential of nectin-2 to interact with HHV-6B gB in both prefusion and postfusion forms. And while DI lies relatively close to the membrane in both conformations, the unresolved membrane proximal region of gB and the additional Ig-like domains in full-length nectin-2 likely provide additional spatial extension on both sides, allowing receptor access without membrane obstruction (Fig 5B). Furthermore, given the dynamic nature of gB transitions, it remains possible that nectin-2 engages HHV-6B gB in multiple conformational states, which may allow for greater spatial flexibility for interaction.

In summary, our AlphaFold2-based structural modeling suggests that the interaction between HHV-6B gB and nectin-2 involves specific conserved residues within DI. Moreover, the accessibility of the nectin-2 binding site in both the prefusion and postfusion conformations of HHV-6B gB supports a possible role for nectin-2 in facilitating viral infection by interacting with both gB conformations.

## Discussion

As a member of the β-herpesvirus subfamily, HHV-6B primarily infects infants and young children, with a high prevalence in the general population [4]. HHV-6B has been implicated in various diseases, especially those that pose a serious threat to immunocompromised individuals such as transplant recipients [47]. It has also recently been shown to reactivate during

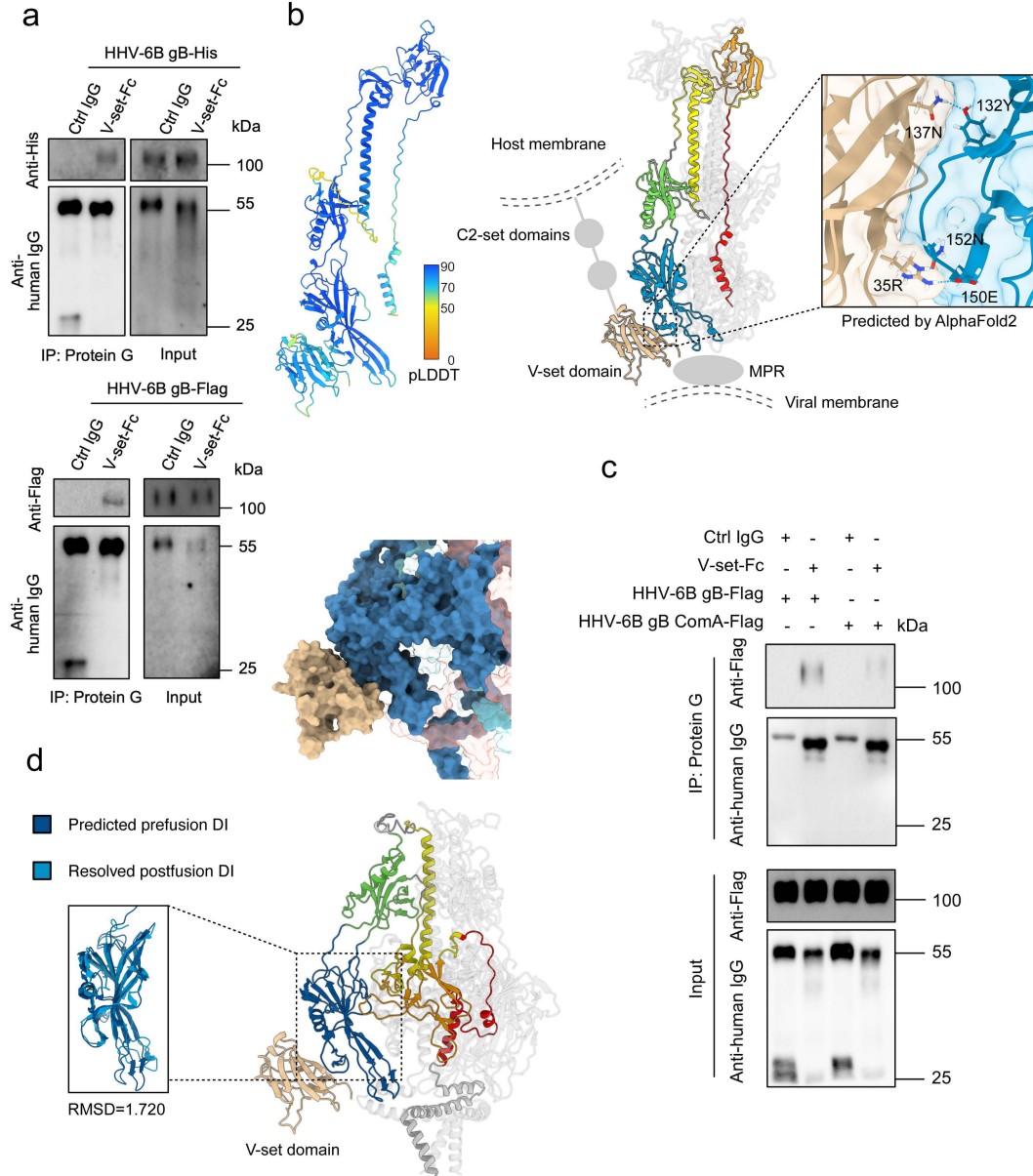

**Fig 5. Structural modeling of the interaction of HHV-6B gB with its receptor nectin-2. a**, Immunoprecipitation of HHV-6B gB with V-set-Fc. V-set-Fc or irrelevant human IgG (control) was incubated with purified Flag- or His-tagged HHV-6B gB. Immunoprecipitation was performed using protein G beads, followed by SDS–PAGE and immunoblotting. The interaction between V-set-Fc and HHV-6B gB was detected using anti-human IgG and anti-Flag or anti-His antibodies. **b**, Predicted interaction between the nectin-2 V-set domain and HHV-6B gB. The interaction between the nectin-2 V-set domain and HHV-6B gB was predicted via AlphaFold2 based on the resolved HHV-6B gB structure and the reported V-set structure (PDB: 3R0N). The pLDDT confidence score for the prediction is indicated by the scale bar. The central panel displays a cartoon representation of the predicted interaction, with the HHV-6B gB protomer colored as in Fig 1B. A zoomed-in view highlights the predicted interacting residues, shown as sticks. To illustrate the spatial context of this interaction relative to the membranes, the approximate positions of the unresolved membrane-proximal region (MPR) of gB, the additional Ig-like C2 domains of full-length nectin-2, and the viral and host membranes are schematically indicated. **c**, Immunoprecipitation of wild-type and mutant HHV-6B gB with V-set-Fc. Wild-type HHV-6B gB-Flag and ComA-Flag (mutant with Y132A, E150A, and N152A substitutions) were tested for interaction with V-set-Fc or control IgG. Binding was assessed by immunoprecipitation and immunoblotting as in (a). **d**, Modeling of prefusion HHV-6B gB and its interaction with V-set. The prefusion model of HHV-6B gB was generated via SWISS-MODEL based on the resolved prefusion structure of HCMV gB (PDB ID: 7KDP). The modeled DI in prefusion conformation and the resolved DI in postfusion conformation of HHV-6B gB were compared via ChimeraX v1.6, and the RMSD value is displayed. The modeled DI in prefusion conformation was aligned with the predicted nectin-2 V-set domain from the postfusion HHV-6B gB-nectin-2 complex. The interaction is depicted with the gB-nectin-2 complex shown in cartoon form, while the interaction interface is highlighted using surface representation.

CAR-T-cell production, potentially leading to severe complications such as encephalitis [9]. Despite its high infection rate and associated health risks, there are still no vaccines or specific antiviral therapies targeting HHV-6B.

gB is essential for herpesvirus infection and is a key target for neutralizing antibodies (nAbs) and vaccine development [10,48,49]. Several nAbs targeting gB have been identified against different herpesviruses, with the humanized anti-HSV gB antibody HDIT101 advancing to phase II clinical trials [50]. Additionally, vaccines incorporating gB as an immunogen for HSV, HCMV, and EBV have entered clinical trials. Notably, previous vaccine candidates have typically used recombinant postfusion gB ectodomains due to the instability of the prefusion conformation, and it has been assumed that prefusion gB would elicit superior neutralizing responses. However, a recent study on HCMV gB successfully stabilized its prefusion conformation through introducing mutations, but found it did not induce higher neutralizing antibody titers in mice compared to postfusion gB, suggesting that both pre- and postfusion gB conformations warrant consideration for vaccine design [51].

In this study, we resolved the cryo-EM structure of the HHV-6B gB ectodomain in postfusion conformation, revealing a typical trimeric spike-like conformation characteristic of postfusion herpesvirus gB, as well as unique features. Notably, the N-terminal region of HHV-6B gB is largely structured and resolved starting from residue 25, unlike the typically unresolved N-termini observed in other herpesvirus gB. Sequence alignment revealed that HHV-6B gB has a notably shorter N-terminal region than other herpesvirus gB do. Structurally, the N-terminus residues $^{27}$YIT$^{29}$ form a β-strand that contributes to the formation of DIV and is located on the exterior of DIV, whereas in other gB, the corresponding β-strand is typically positioned near the central axis above DIII. Previous studies have shown that the unstructured N-terminus of gB, which is largely absent in HHV-6B, could be functionally important for other herpesviruses. For example, insertion mutations in the N-terminus of EBV gB can significantly impair its fusion activity, whereas N-terminal $^{109}$KSQD$^{112}$ mutation can greatly reduce VZV propagation [21]. Additionally, the N-terminal functional region 4 (FR-4) of HSV-1 gB and the N-terminal antigenic domain 2 (AD-2) of HCMV gB were reported to be important nAb targets [52–54]. Interestingly, a recent study described the prefusion-stabilized structures of multiple herpesvirus gBs, in which portions of the N-terminal regions were observed [55]. These findings revealed that gB N-terminus adopt two main interaction patterns: binding either to DIV or to the DI hinge. Among them, the N-terminus of VZV gB was resolved in the most detail, showing clear interactions with the DI hinge, which suggests a potential role in stabilizing the prefusion state and regulating fusion activation. However, this structural feature was not observed in EBV gB, indicating that not all herpesvirus gBs utilize their N-terminal regions in the same way. In the case of HHV-6B gB, its unique short N-terminus and the distinct position of its first β-strand may lead to functional and immunogenic differences. This also raises the question of whether parts of the DIV in HHV-6B gB may have assumed functions typically associated with the unstructured N-terminus in other herpesviruses, thereby making DIV a more functionally and immunologically critical domain.

Moreover, we have, for the first time, resolved the structure of the furin site in gB. The furin site, characterized by the proteolytic motif R-X-K|R-R, is present in gB from all human herpesviruses except HSV. In previously resolved gB structures, the furin site has not been captured with reliable density, likely owing to the inherent flexibility of this region. Even when mutations were introduced to prevent cleavage, structural resolution remained elusive [19]. In contrast, despite not introducing any mutations, we observed a clear density corresponding to the HHV-6B furin site region, which remains an intact flexible linker containing residues $^{396}$RRRRD$^{400}$. Interestingly, we found that the proportion of cleaved gB in our purified protein was low, and an *in vitro* assay confirmed that the ectodomain of HHV-6B gB is highly resistant to furin cleavage, potentially due to the sequence preferences of furin. Notably, the recombinant HHV-6B gB ectodomain analyzed in our study is expected to rapidly adopt a postfusion conformation upon translation given the absence of the membrane anchor, and whether furin cleavage occurs similarly in full-length, prefusion gB remains to be determined. In some viruses, such as flaviviruses and retroviruses, fusion protein activation is dependent on furin cleavage for functionality [56,57]. In contrast, herpesvirus gB does not appear to strictly require furin cleavage for cell entry, and its functional significance varies across different herpesviruses [23,25,42,43,58–60]. Furin cleavage can generate a CendR motif that is involved in

binding to the b1 domain of NRP1, a reported receptor for EBV and KSHV gB[42, 43]. The furin cleavage motif in VZV gB contributes to pathogenesis in skin tissue *in vivo* [23]. Besides, in PrV, the furin site in gB is essential for syncytium formation and pathogenesis *in vivo* but not for viral entry or replication [60,61]. In this study, we provide structural insights into the HHV-6B gB furin site; however, its functional role in viral entry, infectivity, or pathogenesis remains to be elucidated. Our resolved structure of the furin site provides a structural foundation for future studies on the varying functions of the furin sites of HHV-6B and other herpesvirus gB.

We further examined the similarities and differences between HHV-6B gB and other herpesvirus gB. Our analysis revealed that the amino acid variations between HHV-6A and HHV-6B are predominantly located in the exposed DI, DII, and DIV of gB, whereas the DIII and DV cores are highly conserved, a pattern also observed in HCMV strains [19]. Furthermore, we also observed a general trend across herpesvirus gB wherein highly conserved residues tended to cluster within internal regions. This likely reflects the adaptation of exposed regions to receptor interactions and immune pressures, while core domains are preserved for structural stability, as is the case for the SARS-CoV-2 spike protein [62]. Further investigations are needed to clarify the functional implications of these variations in HHV-6B. In addition, we found that key structural elements, the 10 cysteine residues that form intrachain disulfide bonds, are fully conserved across these viruses, underscoring their shared structural importance. Despite the overall structural similarity across gB, we observed variations in the lengths and folding angles of certain structural elements, as well as differences in the spatial orientation within the overall gB architecture.

We then explored how these structural similarities and differences might affect the vulnerability of HHV-6B gB. In the absence of HHV-6B-specific nAbs, we investigated whether nAbs targeting gB from other herpesviruses could cross-react with HHV-6B gB and examined the conservation of their binding epitopes. BLI revealed no cross-reactivity between these nAbs and HHV-6B gB. When their footprints were aligned with HHV-6B gB, we identified a few conserved residues within the binding region and structural similarity in domains, especially in DII and DIV, although most interactions varied significantly. Notably, none of the nAb footprints aligned with residues universally conserved across herpesvirus gB, likely due to the internal positioning of these conserved residues, which makes them less accessible for antibody binding. Among the six nAbs, the epitope of SM5–1, which binds to HCMV gB DII, exhibited the highest degree of conservation with the corresponding region on HHV-6B gB, with two of the six interacting residues matched. The differences in the epitopes underscore the unique vulnerability of HHV-6B gB. Furthermore, we observed that certain nAb footprints from different herpesviruses aligned to overlapping regions on HHV-6B gB. For instance, the footprints of HDIT102, targeting HSV-1 DI, and 1G2, targeting HCMV DI, overlapped at residues $G^{229}$, $S^{230}$ and $I^{244}$ on HHV-6B gB. Similarly, the footprints of 93K, targeting VZV DIV, and 3A5, targeting EBV DIV, overlapped at residues $K^{494}$ and $V^{527}$. These aligned regions recognized by different nAbs may represent functionally important cross-vulnerable sites, which could serve as important targets for neutralization in HHV-6B.

The role of gB as a target for nAbs is based on its essential function in viral infection. In addition to mediating membrane fusion through conformational changes, a universally essential step for herpesvirus entry into various cell types, gB also contributes to viral tropism by engaging specific cellular receptors. For example, both EBV and KSHV gB bind NRP1, activating downstream signaling pathways that promote internalization and infection in nasopharyngeal epithelial cells and mesenchymal stem cells, respectively [42,43]. Similarly, HSV-1 gB interacts with receptors such as PILRα, NMHC-IIA, and MAG, facilitating viral entry into diverse cell types [63]. Nectin-2 was the first molecule identified to mediate HHV-6B infection through its interaction with gB. Although nectin-2 is less efficient at promoting viral entry than the CD134 receptor, which is predominantly expressed on activated T cells, nectin-2 is more widely expressed across various cell types permissive to HHV-6B infection and may play a critical role in the pathogenesis of HHV-6B-associated diseases in the liver and central nervous system [15]. In this study, we confirmed this interaction through an immunoprecipitation assay with purified proteins and used AlphaFold2 to predict the interaction. These results suggest that nectin-2 binds to the external part of the base of HHV-6B gB DI. While the predicted binding sites on DI are conserved across the HHV-6A and HHV-6B

strains, nearby residues show species-specific differences. As salivary gland cells are also permissive for HHV-6A, it remains to be investigated whether HHV-6A also utilizes nectin-2 for infection. To further explore potential binding in different conformations, we used SWISS-MODEL to generate a prefusion HHV-6B gB model on the basis of the resolved prefusion HCMV gB template. The model indicates that the nectin-2 binding site on DI remains exposed and retains a fold similar to that of the postfusion conformation. This suggests that nectin-2 potentially interacts with HHV-6B gB in both conformations, which may either facilitate viral attachment or assist in the process of membrane fusion, though this remains speculative and requires further validation.

In summary, this study provides important insights into the structure, vulnerabilities and receptor-binding mechanisms of HHV-6B gB. The resolution of key structural features, including the unique N-terminal region and the conserved furin site, along with our investigation of its interaction with the receptor nectin-2, significantly advances our understanding of HHV-6B gB's functional role and its involvement in the viral infection process. Moreover, our parallel neutralizing epitope comparison revealed distinct vulnerability of HHV-6B gB while also highlighting potential neutralization targets. Collectively, these findings enhance our understanding of HHV-6B infection mechanisms and will inform future efforts to develop targeted antibodies or vaccines.

## Methods

### Plasmid construction

HHV-6B gB (residues 1–680, strain Z29, UniProtKB accession number: P36320) was subjected to codon optimization for expression in 293F cells. The optimized sequence was synthesized and cloned and inserted into a pCAGGS vector containing a Kozak sequence (GCCACC) upstream of the start codon, with a 6*His-tag or Flag-tag (DYKDDDK) incorporated at the C-terminus prior to the stop codon. Fusion loop substitutions (VGVV$^{102\text{-}105}$ to HRTT and WLY$^{187\text{-}189}$ to ATH) were then introduced via homologous recombination. Primers containing the desired mutations, along with flanking homology arms, were synthesized and used to amplify DNA fragments incorporating the mutated sites. The wild-type HHV-6B gB (residues 1–680, tagged with 6*His or Flag) plasmid was used as the template, and the amplified fragments were then cloned and inserted into a pCAGGS vector digested with EcoRI (NEB, CAT#R3101V) and MluI (NEB, CAT#R3198V) via the ClonExpress Ultra One Step Cloning Kit V2 (Vazyme, CAT#C116-02). The resulting plasmid, pCAGGS-HHV-6B-gB-His or pCKAGGS-HHV-6B-gB-Flag, was used for further studies.

Human furin (residues 23–574, UniProtKB accession number: P09958), supplemented with an N-terminal CD5 signal peptide, was subjected to codon optimization for expression in 293F cells. The optimized sequence was synthesized and cloned and inserted into a pCAGGS vector containing a Kozak sequence upstream of the start codon, with a 6*His-tag incorporated at the C-terminus prior to the stop codon. The resulting plasmid, pCAGGS-Furin-His, was used for further studies.

The human nectin-2 V-set domain (residues 1–158, UniProtKB accession number: Q92692), fused at the N-terminus with the Fc region of human IgG, was subjected to codon optimization for expression in 293F cells, synthesized and cloned and inserted into a pCAGGS vector containing a Kozak sequence upstream of the start codon. The resulting plasmid, pCAGGS-V-set-Fc, was used for further studies.

### Protein expression and purification

For protein expression, Expi293F cells were cultured to a density of $1.5 \times 10^6$ cells/mL and transiently transfected with the respective plasmids mentioned above for HHV-6B gB-His, HHV-6B gB-Flag, V-set-Fc, and Furin-His expression. The transfection mixture consisted of 1 mg of plasmid and 3 mg of polyethyleneimine linear (PEI) MW40000 (Yeasen, CAT#40816ES03) per 800 mL of culture medium. The supernatant was collected by centrifugation at $8,000 \times g$ for 1 hour at 4°C five days posttransfection. The supernatant was then filtered through a 0.45 µm vacuum-driven filter (Cat# XBXY-51), and the following purification methods were used:

For HHV-6B gB-His, the filtered supernatant was incubated with Ni Sepharose resin (Cytiva, Cat# 17371202) for immobilized metal affinity chromatography (IMAC). After washing with 10 column volumes of PBS, the protein was eluted with 300 mM imidazole in PBS. The eluted fractions were concentrated to 1 mL with a 100K MWCO filter (Millipore, CAT#UFC9100), clarified by centrifugation at 18000 × g for 10 minutes at 4°C, and further purified via a Superdex 200 Increase 10/300 GL column (Cytiva, Cat#28990944) on an AKTA Pure25m system, with PBS as buffer. Fractions containing HHV-6B gB-His were collected, concentrated, and buffer-exchanged into PBS with 0.1% DDM (n-dodecyl β-D-maltoside; Macklin, Cat#69227-93-6). The purified protein was then prepared for structural analysis.

For HHV-6B gB-Flag or HHV-6B gB ComA-Flag (harboring Y132A, E150A, and N152A substitutions), the supernatants were incubated with Flag beads (Sigma–Aldrich, CAT#A2220-3ML), followed by washing with 10 column volumes of PBS. The proteins were eluted with 150 μg/mL 3xFlag peptide (Beyotime, Cat# P9801-25 mg) dissolved in PBS. The eluates were concentrated to 1 mL with a 100K MWCO filter (Millipore, CAT#UFC9100), clarified by centrifugation at 18000 × g for 10 minutes at 4°C, and further purified via a Superdex 200 Increase 10/300 GL column (Cytiva, Cat#28990944) with PBS as buffer. Fractions containing the target proteins were collected, concentrated and stored at -80°C.

For the Furin-His protein, the purification of Furin-His followed the same IMAC procedure as that used for HHV-6B gB-His, with a 10K MWCO filter (Millipore, Cat# UFC901096–24EA) used for the concentration step. After SEC, the protein was concentrated and stored at -80°C.

For the V-set-Fc protein, the filtered supernatant was incubated with protein G resin (Cytiva, Cat# 17061805), and after washing with 10 column volumes of PBS, the protein was eluted with 0.2 M glycine at pH 3. The eluted protein was immediately neutralized with 1.5 M Tris-HCl (pH 8.8; Beyotime, Cat#ST789–500ml), concentrated with a 30K MWCO filter (Millipore, Cat#UFC9030), and purified via SEC using PBS as the buffer. The protein was concentrated and stored at -80°C.

For the purification of other herpesvirus gB (HSV-1, VZV, HCMV, EBV, and KSHV gB), the ectodomain was similarly engineered with a His-tag at the C-terminus, and fusion loops were mutated as previously described in related studies. Their purification followed the same IMAC procedure as that used for HHV-6B gB-His. After purification by SEC, the proteins were concentrated and stored at -80°C. For the anti-gB antibodies used in this study (HDIT102, 1G2, SM5–1, 93K, 3A3, and 3A5), the purification procedure followed the same method as that used for V-set-Fc. The purified antibodies were stored at -80°C.

## Cryo-EM sample preparation and data collection

To image HHV-6B gB by cryo-EM, a 4 μL sample with a concentration of 10 mg/ml was loaded onto freshly glow-discharged (45 s at 15 mA) Quantifoil Au R1.2/1.3 300 mesh grids. Plunge freezing was performed via a Vitrobot Mark IV system (Thermo Fisher Scientific, Inc.), using a 10 s wait time, 0 blot force, 5 s blot time, 100% humidity and 6°C. Cryo-EM data were collected on a 300 kV Titan Krios microscope (Thermo Fisher Scientific, Inc.) equipped with a BioContinuum Imaging Filter (Gatan, Inc.). The images were recorded on a K3 Summit direct detection camera (Gatan Inc.) via EPU software and the GIF Quantum energy filter for automated image acquisition. The images were recorded at a nominal magnification of 130,000 × in superresolution mode, yielding a calibrated pixel size of 0.66 Å. Each exposure was dose-fractionated into 32 frames, leading to a total dose of 50 e⁻/Å². The final defocus range of the micrographs was -2.0 to -1.0 μm.

## Cryo-EM data processing

All image processing was performed in cryoSPARC (v3.3.2) [64]. The raw movie frames were dose weighted and corrected for beam-induced drift via patch motion correction (multi) integrated in cryoSPARC. The contrast transfer function (CTF) parameters were estimated via patch CTF estimation. The particles were first picked via a blob picker. The particle images were extracted via a box size of 100 pixels (bin 4) for two-dimensional classification. High-quality particles were selected for Topaz training, and another round of particle picking was performed. Prior to building the initial model, rebalancing 2D classification was required to obtain similar particle numbers representative of projections in different

orientations. After ab initio model building and heterogeneous refinement, good class particles were selected for the construction of the new initial model. Heterogeneous refinement and homogeneous refinement were carried out, and specific class particles were utilized to generate the final map via nonuniform (NU) refinement (to which C3 symmetry was applied) with a global resolution of 2.8 Å [65]. The reported resolutions are based on the gold-standard FSC criterion of 0.143.

### Model building, refinement, and validation

The structure of the HHV-6B gB protein (predicted by Alpha Fold2) was docked into the cryo-EM density maps via CHIMERA [66–68]. The models were manually corrected for local fit in COOT [69]. The models were refined against corresponding maps in real space via PHENIX [70], in which secondary structural restraints and Ramachandran restraints were applied. The stereochemical quality of each model was assessed via MolProbity. The statistics for model refinement and validation are shown in S2 Fig and S1 Table.

### AlphaFold3 prediction and structural comparison

The structure of the HHV-6B gB ectodomain (residues corresponding to the experimentally resolved region) was predicted using AlphaFold3 via the public web server (https://alphafoldserver.com/) [71]. The prediction was performed using default parameters. Confidence in the predicted structure was assessed using per-residue pLDDT scores. The highest-ranked model was used for structural alignment and visualization using PyMOL.

### Multiple sequence alignment

To evaluate the sequence similarity across human herpesviruses, protein sequences were retrieved from UniProtKB, including gB sequences from HHV-6B (strain Z29, accession P36320), PrV (strain Kaplan, accession G3G8X1), HSV-1 (strain KOS, accession P06437), VZV (strain Oka, accession Q4JR05), HCMV (strain AD169, accession P06473), EBV (strain B95-8, accession P03188) and KSHV (strain GK18, accession F5HB81). MSA was performed via ClustalW (https://www.genome.jp/tools-bin/clustalw), and the resulting alignment was visualized with ESpript3.X (https://espript.ibcp.fr/ESPript/cgi-bin/ESPript.cgi) for clearer representation of conserved regions [72]. Manual adjustment was performed to ensure conserved cysteines are precisely aligned.

To investigate interspecies and intraspecies variations in gB amino acid sequences between HHV-6A and HHV-6B, we retrieved gB sequences for each virus strain from the NCBI database. MSA was performed via ClustalW (https://www.genome.jp/tools-bin/clustalw), and the resulting alignment was visualized with ESpript3.X (https://espript.ibcp.fr/ESPript/cgi-bin/ESPript.cgi). On the basis of the alignment results, we categorized the amino acid residues from positions 20–680 into three groups: (1) conserved across all analyzed strains, (2) conserved within HHV-6B strains, and (3) variable among HHV-6B strains.

### Sequence conservation analysis and mapping

The MSA result of multiple herpesvirus gB mentioned above in a Clustal format file was used to calculate sequence conservation for each position via the AL2CO program (https://github.com/TheApacheCats/al2co.git), which computes positional conservation scores [73]. The resulting sequence conservation was visualized and graphed via ggplot2. To map sequence conservation onto the HHV-6B gB structure, ChimeraX v1.6 was used, which integrates AL2CO scores with the resolved structure.

### Prediction of the furin cleavage potential

The sequences of the human herpesvirus gB used in the h were retrieved from UniProtKB and analyzed for furin cleavage potential via ProP-1.0 (https://services.healthtech.dtu.dk/services/ProP-1.0/). The tool was used to predict cleavage potential scores and corresponding cleavage sites of each gB ectodomain. The results were visualized as bar graphs via GraphPad Prism.

### *In vitro* furin cleavage assay

Purified HHV-6B gB-His and KSHV gB-His were first buffer-exchanged into cleavage reaction buffer (20 mM HEPES, 0.1% Triton X-100, and 1 mM $CaCl_2$; pH 7.5) via ultrafiltration with 100K MWCO filters (Millipore, CAT#UFC9100) to remove the PBS. After buffer exchange, the proteins were divided into aliquots of 20, 50, or 100 µg. Each aliquot was incubated with 5 µg of purified furin in a 100 µL reaction mixture. Controls with 50 µg of HHV-6B gB-His or KSHV gB-His protein without furin were also prepared. The reactions were conducted at 37°C for 3 hours. The samples were prepared with 5×SDS–PAGE Plus Sample Buffer (Genstar, CAT#E151-10), which contains reducing agents, followed by SDS–PAGE and Coomassie blue staining.

### Biolayer interferometry

To determine the cross-reactivity of gB nAbs from different herpesviruses with HHV-6B gB, a biolayer interferometry (BLI) assay was performed via protein A biosensors (Sartorius, CAT#18–5010). The nAbs HDIT102, 1G2, SM5–1, 93K, 3A3, and 3A5 were captured at 10 µg/mL after equilibration in kinetic buffer, which was PBS containing 0.01% Tween-20. The sensors with bound antibodies were then exposed to serial dilutions of HHV-6B gB-His (800, 400, 200, 100, and 50 nM), plus a 0 nM baseline control, for an association phase of 150 s, followed by dissociation in kinetic buffer for 150 s.

After regeneration and neutralization via 0.2 M glycine (pH 3) and kinetic buffer, respectively, sensors were reloaded with the antibodies and tested against their respective target gB, which were HDIT102 for HSV-1 gB, SM5–1 and 1G2 for HCMV gB, 93K for VZV gB, and 3A3 and 3A5 for EBV gB. Raw curves were aligned to the 0 nM baseline, and kinetic parameters were calculated by fitting the data to a 1:1 binding model. The results were visualized via GraphPad Prism.

### Footprint analysis and identification of residues involved in antibody-gB interactions

The interaction footprints of antibodies on gB were determined via the Rosetta suite, which incorporates the following gB-antibody complex structures: PDB: 8RH0, 6VN1, 5C6T, 7KDD, and 7FBI. The InterfaceAnalyzer tool within Rosetta was utilized to define the antibody footprints on gB, with a surface probe radius set to 1.4 Å and a protein–protein interaction cutoff radius of 6 Å. Additionally, specific contact residues were identified via PDBePISA (https://www.ebi.ac.uk/msd-srv/prot_int/pistart.html).

### Prediction of the interaction between HHV-6B gB and the nectin-2 V-set domain via AlphaFold2

To predict the interaction between HHV-6B gB and the nectin-2 V-set domain, AlphaFold2 (https://colab.research.google.com/github/sokrypton/ColabFold/blob/main/AlphaFold2.ipynb) was employed [74]. Specifically, custom templates were created using the resolved HHV-6B gB structure in this study and the reported structure of the nectin-2 V-set domain (PDB: 3R0N) [45]. The amino acid sequences of both gB and nectin-2 were input into the system for multimer prediction. The model that ranked first on the basis of confidence scores was selected as the final predicted structure for further analysis.

### Homology modeling of the HHV-6B gB prefusion structure and nectin-2 interaction prediction

The full-length sequence of HHV-6B gB (strain Z29, UniProtKB accession number: P36320) was used as input for homology modeling via SWISS-MODEL [75–79]. The resolved structure of HCMV gB in the prefusion conformation (PDB ID: 7KDP) was selected as the template [46]. The model was built on the basis of sequence alignment between HHV-6B gB and HCMV gB, ensuring the conservation of key structural elements. Next, DI of the prefusion HHV-6B gB model was aligned with the predicted postfusion HHV-6B gB-nectin-2 V-set complex, which was previously generated via AlphaFold2, to map potential nectin-2 binding regions in the prefusion conformation. This alignment was performed via PyMOL. The newly aligned prefusion gB-nectin-2 complex was then refined and optimized via Rosetta to produce a more stable and realistic structural model of the interaction.

## Immunoprecipitation assay

Four micrograms of V-set-Fc or 4 µg of irrelevant human IgG as a control was incubated with 8 µg of purified HHV-6B gB-His at 4°C for 2 hours with gentle rotation. After incubation, a portion was reserved as input, and protein G beads (Cytiva, Cat# 17061805) were added to the remaining samples, which were subsequently incubated overnight at 4°C. The beads were washed six times with buffer containing 50 mM HEPES (pH 7.4), 150 mM NaCl, 5 mM EDTA, and 0.5% NP40. The samples were then prepared via 5×SDS–PAGE Plus Sample Buffer (Genstar, Cat# E151-10) and boiled for 5 minutes at 100°C. Immunoblotting was performed using anti-human IgG (Abcam, Cat# ab109489–40ul), anti-His (CST, Cat# 12698S), and anti-rabbit antibodies (Thermo Fisher, Cat# 31460). For an additional experiment involving HHV-6B gB-Flag or HHV-6B gB ComA-Flag, the same protocol was followed, with the anti-Flag antibody (CST, Cat#14793S) replacing the anti-His antibody for detection in the immunoblotting step.

## Supporting information

**S1 Fig. Purification of the HHV-6B gB680-mut ectodomain. a**, Size-exclusion chromatography (SEC) profile for HHV-6B gB680-mut ectodomain, performed on a Superdex 200 Increase 10/300 GL column. **b**, SDS-PAGE analysis showing the purification results for both wild-type HHV-6B gB680 (HHV-6B gB680-wt) and the mutant construct (HHV-6B gB680-mut).
(TIF)

**S2 Fig. Cryo-EM workflow for data collection, processing, and model building of the HHV-6B gB ectodomain.** The workflow includes motion correction, contrast transfer function (CTF) estimation, particle picking, 2D classification, *ab initio* 3D reconstruction, and iterative heterogenous refinement steps. The Gold-standard Fourier shell correlation curve for the HHV-6B gB ectodomain. The Gold-standard Fourier shell correlation (FSC) curve for the HHV-6B gB ectodomain indicates a final refined resolution of 3.0 Å. The structure model is presented at the bottom of the figure, with both the FSC curve and the map of orientational distribution displayed to its right.
(TIF)

**S3 Fig. Comparison with AlphaFold3 prediction and domain details of HHV-6B gB. a, AlphaFold3-predicted structure colored by pLDDT confidence score (left) and structural alignment of the AlphaFold3 model with the cryo-EM structure (right).** Zoomed-in views of selected regions highlight structural differences, including the DII–DIII linker containing the furin cleavage site and the fusion loop region. The table summarizes the RMSD values (in Å) between the predicted and resolved structures for individual domains and the ectodomain. **b,** The HHV-6B gB monomer is shown in cartoon form, with each domain colored as described in Fig 1C. The starting and ending residues of each domain segment are labeled. The fusion loops within DI are highlighted in red, with the mutated and wild-type amino acid sequences annotated for comparison. The furin cleavage site in the linker region between DII and DIII is colored pink, with the corresponding amino acid sequence and the precise furin cleavage site indicated. In the enlarged view of domain DII, the dashed segment marked by a pink asterisk indicates the unresolved region corresponding to residues 384–390 in the cryo-EM structure.
(TIF)

**S4 Fig. Comparative sequence analysis of HHV-6B gB and other herpesvirus gB.** Multiple sequence alignment of HHV-6B gB with gB from HSV-1, VZV, HCMV, EBV, and KSHV was performed, including sequences of gB from HHV-6B (strain Z29), PrV (strain Kaplan), HSV-1 (strain KOS), VZV (strain Oka), HCMV (strain AD169), EBV (strain B95-8), and KSHV (strain GK18). Amino acids that are conserved across all seven gB are highlighted with red boxes and white text, while similar residues are shown in blue boxes with red text. The structural domains of HHV-6B gB are displayed above the alignment, colored following the scheme shown in Fig 1C. The first N-terminal residue resolved in the structures

shown in Fig 2A is marked with a blue box, and the first N-terminal beta-strand is outlined with an orange box. Furin cleavage sites are marked with pink boxes. The two α-helices in the DV of each gB are marked with green lines beneath the sequences. The ten cysteine residues involved in disulfide bond formation are indicated with green dots above the aligned sequences.
(TIF)

**S5 Fig. Furin cleavage potential and experimental cleavage analysis of HHV-6B gB. a**, Furin cleavage potential was predicted via ProP-1.0, based on gB extracellular sequences of HHV-6B (strain Z29, accession P36320), HSV-1 (strain KOS, accession P06437), VZV (strain Oka, accession Q4JR05), HCMV (strain AD169, accession P06473), EBV (strain B95-8, accession P03188), and KSHV (strain GK18, accession F5HB81). The cutoff score, 0.5, indicates the presence of potential furin cleavage sites. **b**, SDS-PAGE analysis of purified HHV-6B gB ectodomains with wild-type or chimeric furin sites. The wild-type furin site (VNLRRRR|DL) was replaced with that of VZV (RNTRSRR|SV) or EBV (VLRRRRR|DA). All proteins were purified using the same method as the original HHV-6B gB680-mut construct. Blue arrow indicates the furin-cleaved fragment.
(TIF)

**S6 Fig. Structural comparisons of herpesvirus gB. a**, Conserved amino acids across HHV-6B, PrV, HSV-1, VZV, HCMV, EBV, and KSHV gB, as identified in S4 Fig, are mapped onto the resolved structure of HHV-6B gB. The HHV-6B gB is shown as a semi-transparent surface model, colored according to Fig 1C, with the conserved residues represented as opaque white spheres to illustrate the relative internal distribution of conserved amino acids within the gB structure. The side, top and bottom views are displayed. **b**, The resolved HHV-6B gB structure was aligned with gB from PrV (PDB: 6ESC), HSV-1 (PDB: 2GUM), VZV (PDB: 6VLK), HCMV (PDB: 5CXF), and EBV (PDB: 3FVC), and RMSD values (in Å) were calculated and displayed for each domain (DI–DV) and the ectodomain. For DI–DV, red shading indicates higher RMSD values and white indicates lower values. For the ectodomain column, green shading indicates lower RMSD values and white indicates higher values. **c**, Structural comparison of the two α-helices in DV from HHV-6B, PrV, HSV-1, VZV, HCMV, and EBV. The helices are depicted as cartoon, with the number of amino acids forming each helix and the angles between them displayed to illustrate structural differences.
(TIF)

**S7 Fig. Cross-reactivity of herpesvirus gB neutralizing antibodies with HHV-6B gB.** Biolayer interferometry (BLI) analysis was performed to evaluate the cross-reactivity of gB-specific neutralizing antibodies from various herpesviruses with HHV-6B gB. The antibodies tested included HDIT102 for HSV-1 gB, SM5–1 and 1G2 for HCMV gB, 93K for VZV gB, and 3A3 and 3A5 for EBV gB. Each antibody was tested for binding both to its respective target gB and to HHV-6B gB. The binding signal of 800, 400, 200, 100, and 50 nM of each gB associating with the corresponding ligand antibody captured on protein A biosensors is shown, followed by the dissociation phase. *KD* represents the equilibrium dissociation constant.
(TIF)

**S8 Fig. Protein purification for immunoprecipitation. a,** Size-exclusion chromatography (SEC) and Coomassie-stained SDS-PAGE analysis under reducing and non-reducing conditions of purified HHV-6B gB-Flag protein. **b,** Size-exclusion chromatography (SEC) and Coomassie-stained SDS-PAGE analysis under reducing and non-reducing conditions of purified V-set-Fc.
(TIF)

**S1 Table. Cryo-EM data collection and validation statistics.**
(DOCX)

**S2 Table. Accession numbers and gB sequences of HHV-6A and HHV-6B strains.**
(XLSX)

## Acknowledgments

We thank all contributors for their support and assistance in this study. We also acknowledge the facilities and resources provided by Sun Yat-sen University Cancer Center and Southern University of Science and Technology.

## Author contributions

**Conceptualization:** Chu Xie, CONG SUN.

**Formal analysis:** Chu Xie, Xin-Yan Fang, Yuan-Tao Liu, CONG SUN.

**Funding acquisition:** Zheng Liu, Mu-Sheng Zeng, CONG SUN.

**Investigation:** Chu Xie, Xin-Yan Fang, Xian-Shu Tian, Hang Zhou.

**Methodology:** Chu Xie, Xin-Yan Fang.

**Project administration:** Chu Xie, Mu-Sheng Zeng, CONG SUN.

**Supervision:** Sen-Fang Sui, Zheng Liu, Mu-Sheng Zeng, CONG SUN.

**Validation:** Xian-Shu Tian, Lan-Yi Zhong, Pei-Huang Wu, Hang Zhou, Peng-Lin Li, Yan-Lin Yang, Zi-Ying Jiang.

**Visualization:** Chu Xie, Xin-Yan Fang, Yuan-Tao Liu.

**Writing – original draft:** Chu Xie, Xian-Shu Tian, CONG SUN.

**Writing – review & editing:** Xin-Yan Fang, Yuan-Tao Liu, Sen-Fang Sui, Zheng Liu, Mu-Sheng Zeng, CONG SUN.

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
