## [Decision Letter · Decision Letter 0]

PPATHOGENS-D-25-00231

Human herpesvirus 6B fusion protein structure, vulnerability mapping and receptor recognition

PLOS Pathogens

Dear Dr. SUN,

Thank you for submitting your manuscript to PLOS Pathogens. Your manuscript has been evaluated by three reviewers who are experts in the field. While the reviewers found the technical aspects of the work to be solid and thought that a structure of an additional gB homolog in its postfusion state was of value, they were concerned that the new structure alone did not advance the knowledge far enough to fully meet the publication criteria of PLOS Pathogens. In particular, the manuscript in its current form provides almost no functional data to support the claims derived from the structural analysis. One reviewer even recommended that the manuscript be rejected. After careful consideration, we would be willing to consider a substantially revised version that should carefully address the reviewers concerns and provide additional functional (or structural) data to support structural insights presented in the current version. Therefore, we invite you to submit a revised version of the manuscript that addresses the points raised during the review process and summarized above.

Please submit your revised manuscript within 90 days Apr 24 2025 11:59PM. If you will need more time than this to complete your revisions, please reply to this message or contact the journal office at plospathogens@plos.org. Please include the following items when submitting your revised manuscript:

We look forward to receiving your revised manuscript.

Kind regards,

Ekaterina E. Heldwein

Academic Editor

PLOS Pathogens

Alison McBride

Section Editor

PLOS Pathogens

 Sumita Bhaduri-McIntosh

Editor-in-Chief

PLOS Pathogens

orcid.org/0000-0003-2946-9497

Michael Malim

Editor-in-Chief

PLOS Pathogens

orcid.org/0000-0002-7699-2064

**Journal Requirements:**

At this stage, the following Authors/Authors require contributions: Chu Xie, Xin-Yan Fang, Yuan-Tao Liu, Xian-Shu Tian, Lan-Yi Zhong, Pei-Huang Wu, Hang Zhou, Peng-Lin Li, Yan-Lin Yang, Zi-Ying Jiang, Sen-Fang Sui, Zheng Liu, Mu-Sheng Zeng, and CONG SUN. Please ensure that the full contributions of each author are acknowledged in the "Add/Edit/Remove Authors" section of our submission form.

2) We noticed that you used the phrase 'unpublished data' in the manuscript. We do not allow these references, as the PLOS data access policy requires that all data be either published with the manuscript or made available in a publicly accessible database. Please amend the supplementary material to include the referenced data or remove the references.

4) We notice that your supplementary Figures, and Tables are included in the manuscript file. Please remove them and upload them with the file type 'Supporting Information'. Please ensure that each Supporting Information file has a legend listed in the manuscript after the references list.

5) In the online submission form, you indicated that "Materials and plasmids are available on request to corresponding authors." All PLOS journals now require all data underlying the findings described in their manuscript to be freely available to other researchers, either

1. In a public repository

2. Within the manuscript itself

3. Uploaded as supplementary information.

**Reviewers' Comments:**

Reviewer's Responses to Questions

**Part I - Summary**

Reviewer #1: In this paper the authors solve a cryo-EM structure of HHV-6B in the post fusion conformation.

Unsurprisingly it looks very similar to other herpesvirus fusion proteins. They can resolve the fuin cleavage site and it appears to be mostly uncleaved.

They authors also compare the structures of other gB proteins with Fabs bound to neutralizing epitopes but no Fabs bound to HHV-6B so this analysis doesn’t add much.

They provide biochemical data that gB binds to Nectin-2 via pulldown assays and model the nectin binding site into post-fusion gB and a hypothetical pre-fusion gB.

Reviewer #2: This paper reports the structure of HHV-6B glycoprotein B in the postfusion conformation at a resolution of 2.8 angstroms as determined by cryo-EM and single particle analysis. Despite the availability of many herpes glycoprotein structures, the fusion mechanism remains unclear. The findings reported in this paper are important and the manuscript is well-written. Please consider the following comments to improve the paper.

Reviewer #3: In this paper, Xie et al. provide an intriguing set of data detailing the structural differences between postfusion gB of HH6B and the gB structures of other Herpesviruses. The study successfully resolved the cryo-EM structure of the HHV-6B gB ectodomain, revealing several unique features, including a well-structured N-terminal region starting from residue 25. In addition, the Furin cleavage site was structurally resolved, providing a first structural description of this functional region found in many gB proteins. The authors also confirmed the interaction between HHV-6B gB and the receptor nectin-2, suggesting this binding could facilitate viral attachment or fusion. Moreover, conserved binding sites on gB were identified that may serve as targets for neutralization and offer insights for the development of targeted therapeutic strategies.

Overall, the study is technically well conducted, although there is almost no functional data that would corroborate the claims and assumptions derived from the determined structure. Therefore, there are several points the authors should address, before the paper can be considered for publication.

**Part II – Major Issues: Key Experiments Required for Acceptance**

Reviewer #1: Introducing point mutations to confirm the nectin binding site on gB would be helpful in validating the model.

Reviewer #2: 1.The structure of the HHV-6B gB cleavage site is of unclear significance. More context is needed for the reader. The functional significance of furin cleavage does vary among herpes gBs as the authors indicate (Line 679). However, many viral fusion proteins are functionally inactive in the absence of furin cleavage. It is important to note that that this is not the case for any of the herpesvirus gBs. Furin cleavage is not required for gB function. Indeed, the authors find that the HHV-6B cleavage site is poorly utilized. Please also emphasize that the function of furin cleavage of HHV-6B has not been evaluated.

2.Extended Fig 8c. The identification of the receptor-binding site on gB is not conclusive. Mutation of predicted gB-nectin-2 interaction sites decreases nectin-2 binding to an undetermined extent, but does not eliminate binding. This gB mutant is not characterized sufficiently. Do the three point mutations cause the protein to globally misfold, what is the effect of these mutations on cell surface expression, etc?

3.Fig. 4A is a confirmation of what has already been published in reference 14 and should be removed.

-Line 561: The AlphaFold2 modeling work presented does little to improve the understanding of structural details of gB-receptor interaction.

-Nectin-2 binding to prefusion gB is highly speculative. Experimental evidence is not provided. This interpretation should be limited to the Discussion and omitted from the Abstract.

4.The structure of pseudorabiesvirus gB has been reported PMID: 29046441. This structure should be considered in the analyses presented.

Reviewer #3: 1) The new postfusion structure for HHV6B gB presented here shows a very high resemblance with previously determined structures of other herpesviruses. The authors should prepare an alphafold model of HHV6B and see how it compares with the determined structure. Are there structural features that alphafold was not able to predict that would be functionally or immunologically interesting?

2) In line 241 the authors write: ‘The unique sequence and structural features of the N-terminus of HHV-6B gB may contribute to its distinct functional and immunogenic properties.’ And also, in the discussion that ‘…its unique short N-terminus and the distinct position of its first b-strand may lead to functional and immunogenic differences.’. The position and sequence difference may well account for immunogenic differences, but how this arrangement could contribute a functional difference compared to other gBs should be better described. It would be good if the authors could elaborate more on this, also putting this in context with the recently published pre-print (https://doi.org/10.1101/2024.10.23.619923). For example, would the N-term generally have a prefusion stabilising role that is missing in HHV6B?

3) It would be great to understand the functional role of the furin cleavage site better. One could e.g. replace it with the VZV cleavage site to make it more efficient and compare it to a mutated version that is not cleavable anymore. But I understand that functional studies for this i.e. in a virus background or in a cell-cell fusion assay is not trivial, but could be done, as shown in Ref. 29.

4) For the nAB epitope comparison the authors map the interacting residues of several nABs on HHV6B. Judging from the EM density shown in Fig. 1, the glycosylations on gB should be visible. There are significant differences in the number of glycosylations on gB from different species, which might cover immunologically important epitopes and this should be considered here.

5) The authors have set up a nice BLI assay system to check for interactions of different gBs with nABs. It would be good to report the different affinities of the different nABs. If possible, it would also be good to use the same assay system to determine the binding affinity of nectin-2 to gB and also the interaction mutant ComA.

6) Did the authors try to determine a structure of the gB-nectin2 complex with cryoEM? This would be a great way to verify the prediction.

**Part III – Minor Issues: Editorial and Data Presentation Modifications**

Reviewer #1: (No Response)

Reviewer #2: 1. The title should include a postfusion descriptor.

Reviewer #3: 1) The authors show that the ectodomain of HHV6B gB can be purified using a fusion loop mutant in contrast to the wt protein construct. In ExtData Fig. 1b the two preparations are shown in SDS PAGE. It would be interesting to what degree the protein is cleaved by furin which could done using reducing conditions. There is no loading control, so it is really hard to tell what the samples really are and how much of each was loaded. It might be more informative to also see the expression levels of these constructs e.g. in cell lysates.

2) In HSV-1 gB, DII features an alpha helix (helix X) that was only resolved in a later crystal structure by Cooper et al. (10.1038/s41594-018-0060-6). Is this helix also present in HHV6B and if yes, was it resolved in the structure?

3) Fig. 1 c) It would be nice to indicate the regions that were not modelled in the structure. Fig. 1 d) Please mark the N- and C-terminus with the first/last built residue number as well as where loops are missing, with residue numbers.

4) Fig. 2 – the arrangement of the individual panels is a bit awkward, especially panel d. This should be rearranged in order to group things that go together.

5) Concerning the furin cleavability – the position of the cleavage site on DII and the possibility to resolve it might mean that the site is not fully accessible to the enzyme, as it tightly interacts with the surface of DII. The question is if this would be the same situation in prefusion conformation. The ectodomain construct most probably adopts the postfusion conformation almost instantly after translation due to the missing transmembrane parts and before the posttranslational modifications are done. But in context of the full-length protein, the situation might be different. The authors should comment on this.

6) In ExtData Fig. 5b, for KSHV there seem to be two possible recognition sites, with one having a higher cleavage potential than the other. Does the analysis done in c) then suggest, that the less efficient cleavage site is the one accessible for the enzyme?

7) In ExtData Fig. 5c it might be better to mark the bands with an asterisk or point next to the band, instead of above, to make the position clearer. I don’t understand the difference between ‘uncleaved portion’ and ‘faint uncleaved extracellular portion’. If one compares VZV and EBV, the ratio of cleaved and uncleaved seems similar (as far as I can tell).

8) From line 459 the authors write ‘Although the overall structures of the analyzed domains of different herpesvirus gB and HHV-6B gB are highly similar, especially for DII and DIV, …’ while before they claim that (line 421) ‘variations in DI, DII, and DIV account for the majority of interspecies and intraspecies differences.’ One of the two claims should be revised to keep it consistent.

9) ExtData Fig. 6 – it would be informative to have an overview of the RMSDs between the individual domains. Also, please provide the RMSD values with a unit (Å?).

10) Fig. 3 - Although colour coded, it would be helpful to put the names of the nABs to the corresponding structures in panel a.

11) Line 588: is the number in parenthesis (m = 1.720) the calculated RMSD?

12) The interaction site of Nectin-2 with DI is very close to the membrane. Would the receptor be able to reach that far and so close to the viral membrane?

13) Line 769 – the Uniprot accession number is wrong.

14) ExtData Fig. 8b – the band for the purified V-set-Fc protein runs quite low, as I would expect the size of the monomer to be ~70 kDa. Is that normal?

15) At the end of the discussion the authors claim that their findings will inform future efforts to develop vaccines. This claim, but also implications regarding the vulnerability of gB to neutralising antibodies should be discussed in light of recent findings by Sponholtz et. al, where they found that ‘No gB variants (pre or postfusion) in this study consistently elicited complement-independent neutralizing antibodies in mice.’ (doi.org/10.1073/pnas.2404250121)

PLOS authors have the option to publish the peer review history of their article (what does this mean? ). If published, this will include your full peer review and any attached files.

**Do you want your identity to be public for this peer review?** For information about this choice, including consent withdrawal, please see our Privacy Policy .

Reviewer #1: No

Reviewer #2: No

Reviewer #3: No

**Figure resubmission:**
---

## [Decision Letter · Decision Letter 1]

PPATHOGENS-D-25-00231R1

Human herpesvirus 6B glycoprotein B postfusion structure, vulnerability mapping, and receptor recognition

PLOS Pathogens

Dear Dr. SUN,

Thank you for submitting your manuscript to PLOS Pathogens. After careful consideration, we feel that it has merit but does not fully meet PLOS Pathogens's publication criteria as it currently stands. Therefore, we invite you to submit a revised version of the manuscript that addresses the points raised during the review process.

Please submit your revised manuscript within 30 days Jul 18 2025 11:59PM. If you will need more time than this to complete your revisions, please reply to this message or contact the journal office at plospathogens@plos.org. Please include the following items when submitting your revised manuscript:

We look forward to receiving your revised manuscript.

Kind regards,

Ekaterina E. Heldwein

Academic Editor

PLOS Pathogens

Alison McBride

Section Editor

PLOS Pathogens

Sumita Bhaduri-McIntosh

Editor-in-Chief

PLOS Pathogens

orcid.org/0000-0003-2946-9497

Michael Malim

Editor-in-Chief

PLOS Pathogens

orcid.org/0000-0002-7699-2064

**Additional Editor Comments :**

1. Address the remaining concerns of reviewer 3.

2. Incorporate important functional data currently shown in the Supplement into the main figures, to address the remaining concerns of reviewer 1. For example, Figures S7, S8, S5, and, potentially, other Supplemental figures.

3. Add labels to all the figures so that the reader can follow them without having to go back and forth between the figure and the legend. For example, for each structural models, make sure to indicate whether it is obtained experimentally or predicted in silico. For SEC traces, indicate protein construct, and so on.

**Journal Requirements:**

1) Thank you for stating "Materials and plasmids can be requested from Dr. Cong Sun at suncong@sysucc.org.cn." Please note that your current contact point is a co-author on this manuscript. According to our Data Policy, the contact point must not be an author on the manuscript and must be an institutional contact, ideally not an individual. Please revise your data statement to a non-author institutional point of contact, such as a data access or ethics committee, and send this to us via return email. Please also include contact information for the third party organization, and please include the full citation of where the data can be found.

**Reviewers' Comments:**

Reviewer's Responses to Questions

**Part I - Summary**

Reviewer #1: As per my original review there is very little functional data included in this manuscript, a concern shared by myself and reviewer 3. There is a structure of HHV6 gB and quite a bit of in silico modelling which is hypothesis generating but these are largely untested. There is some additional functional data included in the revision but it is all buried in the supplement. It might be suitable for publication if this were included in the main manuscript but I feel that there is not a significant advance here beyond the structure.

Reviewer #2: The authors did a commendable job in revising the manuscript per the reviewer comments. The manuscript is a valuable contribution to the literature.

Reviewer #3: During the review process Xie et al. put a lot of effort to address all the issues raised by the reviewers. Unfortunately, the functional relevance of the furin cleavage site was not determined. Nevertheless, I think this work and the structural information on HHV-6B gB it provides are relevant and will be interesting for the field of herpes virology. There are still minor things that should be addressed.

**Part II – Major Issues: Key Experiments Required for Acceptance**

Reviewer #1: (No Response)

Reviewer #2: (No Response)

Reviewer #3: (No Response)

**Part III – Minor Issues: Editorial and Data Presentation Modifications**

Reviewer #1: (No Response)

Reviewer #2: (No Response)

Reviewer #3: 1) Comparison of the experimental structure and the alphafold model

It is good that the authors followed my suggestion and performed the alphafold prediction. I think it would be useful to also show the RMSD between the prediction and the experimental structure. The RMSD values could also be incorporated in the table in Extended Data Fig. 6b. These results should also be mentioned in the manuscript, as it is important to show the quality of the prediction, but even more so to also highlight details that cannot be predicted by Alphafold yet. In Extended Data Fig. 3a the differences could be highlighted with zoom-ins on the structural alignment.

2) Functional relevance of the furin cleavage site

I appreciate the effort the authors put in, in order to functionally characterise the furin cleavage during fusion. As the authors pointed out, using HCMV gB enabled fusion, meaning the assay per se is working. This also means that more efficient cleavage of HHV-6B gB (with the EBV / VZV furin cleavage site substitutions) does not increase fusion efficiency of gB. This point might be mentioned in the paper.

3) a-Helix X in HSV-1 gB

This more a comment than a suggestion. I appreciate the authors efforts to clarify the structural differences in the region corresponding to HSV-1 gB residues 462–477. It is indeed interesting that in HHV-6B gB the region forms an extended loop instead of a helix. Also, as far as I can see, the Alphafold prediction shows a loop rather than a helix, further supporting this finding.

4) Nectin-2 gB interaction

I thank the authors for the explanation about the potential interaction. It would be helpful to indicate in Figures 4b and c, where the rest of Nectin-2 would be located. And also, to mention in the text the reasons why a receptor interaction with gB in this position might not be obstructed by the membrane.

PLOS authors have the option to publish the peer review history of their article (what does this mean? ). If published, this will include your full peer review and any attached files.

**Do you want your identity to be public for this peer review?** For information about this choice, including consent withdrawal, please see our Privacy Policy .

Reviewer #1: No

Reviewer #2: No

Reviewer #3: No

**Figure resubmission:**
---

## [Editor Report · Decision Letter 2]

Dear Dr. SUN,

We are pleased to inform you that your manuscript 'Human herpesvirus 6B glycoprotein B postfusion structure, vulnerability mapping, and receptor recognition' has been provisionally accepted for publication in PLOS Pathogens.

Best regards,

Ekaterina E. Heldwein

Academic Editor

PLOS Pathogens

Alison McBride

Section Editor

PLOS Pathogens

Sumita Bhaduri-McIntosh

Editor-in-Chief

PLOS Pathogens

orcid.org/0000-0003-2946-9497

Michael Malim

Editor-in-Chief

PLOS Pathogens

orcid.org/0000-0002-7699-2064
---

## [Editor Report · Acceptance letter]

Dear Dr. SUN,

We are delighted to inform you that your manuscript, "Human herpesvirus 6B glycoprotein B postfusion structure, vulnerability mapping, and receptor recognition," has been formally accepted for publication in PLOS Pathogens.

Best regards,

Sumita Bhaduri-McIntosh

Editor-in-Chief

PLOS Pathogens

orcid.org/0000-0003-2946-9497

Michael Malim

Editor-in-Chief

PLOS Pathogens

orcid.org/0000-0002-7699-2064